

**Potential sources and processes affecting speciated atmospheric mercury at**

**Kejimkujik National Park, Canada**

Xiaohong Xu[1*], Yanying Liao[1], Irene Cheng[2], Leiming Zhang[2*]

[1] Department of Civil and Environmental Engineering, University of Windsor, 401 Sunset

Avenue, Windsor, Ontario, N9B 3P4, Canada

[2]Air Quality Research Division, Science and Technology Branch, Environment and Climate

Change Canada, 4905 Dufferin Street, Toronto, Ontario, M3H 5T4, Canada

Corresponds to Xiaohong Xu (xxu@uwindsor.ca) or Leiming Zhang (leiming.zhang@canada.ca)



**Abstract:** Source apportionment analysis was conducted with Positive Matrix Factorization (PMF) and Principal Component Analysis (PCA) methods using concentrations of speciated mercury (Hg), i.e., gaseous elemental mercury (GEM), gaseous oxidized mercury (GOM), and particulate-bound mercury (PBM), and other air pollutants collected at Kejimkujik National Park, Nova Scotia, Canada in 2009 and 2010. The results were largely consistent between the two years for both methods. The same four source factors were identified in each year using PMF method. In both years, factor Photochemistry and Re-emission had the largest contributions to atmospheric Hg, while the contributions of Combustion Emission and Industrial Sulfur varied slightly between the two years. Four components were extracted with air pollutants only in each year using PCA method. Consistency between the results of PMF and PCA include, 1) most or all PMF factors overlapped with PCA components, 2) both methods suggest strong impact of photochemistry, but little association between ambient Hg and sea salt, 3) shifting of PMF source profiles and source contributions from one year to another was echoed in PCA. Inclusion of meteorological parameters led to identification of an additional component - Hg Wet Deposition in PCA, while it did not affect the identification of other components.

The PMF model performance was comparable in 2009 and 2010. Among the three Hg forms, the agreement between predicted and observed annual mean concentrations were excellent for GEM, very good for PBM and acceptable for GOM. However, on daily basis, the agreement was very good for GEM, but poor for GOM and PBM. Sensitivity tests suggest that increasing sample size by imputation is not effective in improving model performance, while reducing the fraction of concentrations below method detection limit, by either scaling GOM and PBM to higher concentrations or combining them to reactive mercury, is effective. Most of the data treatment options considered had little impact on the source identification/contribution.

## 1. Introduction

Atmospheric mercury (Hg) exists in the form of gaseous elemental Hg (GEM) and oxidized Hg, the latter can be in gaseous phase (gaseous oxidized Hg - GOM) or associated with particulate matter (particulate - bound Hg - PBM). Identification of major sources and processes affecting ambient levels of different Hg forms will help




mitigate the risks of Hg pollution. Atmospheric Hg can be produced from
anthropogenic activities, natural events and re-emission of previously deposited Hg,
the latter two are sometimes grouped together as natural emission sources (Gustin et
al., 2008; Pirrone et al., 2010; UNEP, 2013; Gaffney and Marley, 2014; Zhang et al.,
2016). Natural events consist of volatilization from the ocean, volcanic eruption,
geothermal activities, and weathering of Hg-containing minerals (Pirrone et al., 2010;
Gaffney and Marley, 2014). Small scale or artisanal gold mining, mining and smelting,
and coal combustion are the three major anthropogenic sources (UNEP, 2013; Zhang
et al., 2016). Some of the dry and wet deposited PBM and GOM will be reduced to
GEM in soil, water, and vegetation surfaces where Hg will be re-emitted in the form
of GEM to the atmosphere (Gaffney and Marley, 2014). However, the contributions of
each source and process to a given receptor site are affected by many factors including
proximity to sources and weather conditions.
Various receptor-based models have been used to identify the sources and
processes affecting ambient Hg levels (Cheng et al., 2015). Among these, Positive
Matrix Factorization (PMF) and Principal Component Analysis (PCA) are two
commonly used methods. PMF method provides quantitative source profiles and
source contributions. The resultant source profiles could aid future studies in factor
interpretation. Another strength of PMF is input variable screening and provision of
model performance measures. The users could specify uncertainty values for each
variable in each sample to reduce the impact of measurements with high uncertainties
on the final results (US EPA, 2014a; Hopke, 2016). However, in order to derive
profiles, PMF requires a large number of air pollutants which are often unavailable. In
contrast, PCA can only provide qualitative assessment of sources/processes. One
advantage of PCA over PMF is its capability of allowing inclusion of meteorological
parameters as input, enabling the assessment of the effects of weather conditions on
ambient Hg concentrations (Cheng et al., 2015). Therefore, it is beneficial to conduct
source apportionment of atmospheric Hg using both PMF and PCA. To date, only one
study used this combined approach (Cheng et al., 2009), yet it lacked a thorough
comparison of the results. Furthermore, the ability of receptor models to reproduce the
observed concentrations should be assessed in order to gauge the model performance
(Henry, 1991; Viana et al., 2008), which has been rarely reported in the literature.



The overall objective of this study is to identify the factors affecting ambient Hg concentrations at a receptor site using PMF and PCA approaches. The specific objectives are to, (1) identify the factors affecting ambient Hg concentrations using PCA and PMF model; (2) summarize the similarity and differences in PMF factors and PCA components; (3) evaluate the PMF model performances by Hg forms; (4) investigate the impact of including meteorological parameters on PCA results, and (5) assess the sensitivity of PMF results and performance to different treatment of missing data and low concentration values of speciated Hg.

## 2. Method

### 2.1 Study site

The study site is located in Kejimkujik (KEJ) National Park (44.32°N; 65.2°W; elevation: 170 m), Nova Scotia, Canada. The KEJ site is one of the first speciated Hg sites operated by Environment Canada outside the Arctic. This site was selected primarily because of the bioaccumulation issues at this area. Studies have found that common loons in Kejimkujik National Park had the highest mean blood Hg concentrations in northeastern United States and Southeastern Canada (Evers et al., 2007). Similarly, a 1996/97 survey found that yellow perch and common loons from Kejimkujik National Park and National Historic Site (Nova Scotia) had the highest blood Hg concentrations across North America. A 2006/07 follow up study on yellow perch observed on average a 29% increase in 10 out of 16 lakes, although anthropogenic emission from North America decreased between the mid-90s to the mid-2000s (Wyn, 2010).

The sampling site was surrounded by forests on a flat terrain. It was approximately 50 km away from the nearest coast, 120 km southwest of Halifax, and relatively remote from anthropogenic air emissions. A search of the National Pollutant Release Inventory (NPRI, Environment Canada, 2016) yielded seven Nova Scotia facilities reporting Hg air releases in both 2009 and 2010. Four of them were electric power generation stations, the other three were a refinery, a cement plant, and a university. The provincial annual air emission of Hg were 147.5 kg and 90.3 kg in 2009 and 2010, respectively (Table S1). The two largest Hg emitters were Lingan Power Generating Station (2009-2010 average: 71 kg/yr) and Trenton Power



Generating Station (26 kg/yr), located 450 km and 250 km from the sampling site, respectively. The nearest anthropogenic Hg sources (Dalhousie University, Halifax: 0.17 kg/yr, Imperial oil, Dartmouth Refinery: 2.8 kg/yr) were 140 km northeast of the sampling site. In addition to Hg sources, the nearby NPRI (Environment Canada, 2016) combustion/industrial sources were a biomass-fueled power station and tire production factory located approximately 50 km east/southeast of the KEJ site (Table S1).

**2.2 Data collection**

GEM, GOM and PBM concentrations were collected from 2009 to 2010 using Tekran® instruments (Models 1130/1135/2537) at 3-hour intervals. Hourly concentrations of ground level ozone ($O_3$) and meteorological parameters (temperature, relative humidity, wind speed, and precipitation amount), as well as daily concentrations of $SO_2$ and $HNO_3$, $PM_{2.5}$ (2009 only), and particulate $SO_4^{2-}$, $NO_3^-$, $Mg^{2+}$, $Cl^-$, $K^+$, $Ca^{2+}$, $NH_4^+$, and $Na^+$ were also collected at KEJ site. Detailed information of data collection can be found in Cheng et al. (2013).

Hourly or 3-hr concentrations of GEM, GOM, PBM, $O_3$ and meteorological data were averaged into daily values because PMF and PCA require the same interval for all input variables. All daily values were the same as those used in a PCA study by Cheng et al. (2013). The general statistics of the daily concentrations are listed in Table 1 and Table 2 for year 2009 and year 2010, respectively. The number of missing daily concentrations ranged from 0% (ozone, 2010) to 41% (PBM, 2009), which are excluded from PMF or PCA. Among the three Hg forms, GEM had the fewest values below the Method Detection Limit (MDL), while GOM had the largest percentages of concentrations below MDL, followed by PBM, in both years. The variability, as indicated by coefficient of variability, was low for GEM but much higher for GOM and PBM.

**2.3 Model setup and case design**





Detailed description of the theory of PMF and PCA methods can be found in
Cheng et al. (2015). Model set up and case design are described below.
***PMF***
EPA PMF5.0 (US EPA, 2014b) was used in this study. The 12 cases investigated
are listed in Table 3.   Two approaches were employed in PMF modeling to handle
missing values. The first approach is listwise deletion. Listwise deletion excludes all
the records having one or more missing values, resulting in a complete data matrix as
required in PMF. However, it may cause a large reduction of the dataset when one of
the pollutants has many missing values or several pollutants have missing values at
different time periods. In environmental studies, this approach may lead to biased
results because listwise deletion benefits the records with high concentrations when
below MDL values are flagged as missing (Huang et al., 1999). The second method is
imputation, which increases the sample size in PMF. Hedberg et al. (2005) found that
the relative error of factor profiles deceased as the sample size increased. In this study,
geometric mean and median imputation were used to minimize the undue influence of
extreme values as in Pekey et al. (2004). The effects of imputation was investigated in
Cases 09+Mean, 10+Mean, 09+Median, and 10+Median.
Cases 09+RM, 10+RM, 09-RM, and 10-RM were devised to investigate the
effects of excluding or combining GOM and PBM into reactive mercury (RM) on the
resultant PMF results compared with the full dataset. Uncertainties of GOM and PBM
measurements are considered high (Gustin et al., 2015). It has been reported that
GOM may be collected on the PBM filter thus GOM concentrations could be biased
low (Lynam and Keeler, 2005). Therefore, combining GOM and PBM to RM may
reduce the uncertainties (Cheng et al., 2016). RM was calculated by summing GOM
and PBM when both forms of Hg are detected.
In Case 09ScaleRM and Case 10ScaleRM, a variable scaling factor was used to
increase the GOM and PBM concentrations:
$$\text{scaling factor} = \sqrt{\frac{\max(x)}{x_i}} \qquad\qquad (1)$$

where $x_i$ is the concentration of GOM or PBM in the $i^{th}$ sample. The scaling factor is
large when the concentration is low, and vice versa, but the maximum concentration is





unchanged.
Equation-based uncertainties (US EPA, 2014a) were used in this study, expressed
as:

$$Uncertainty = \frac{5}{6} \times MDL, when\ concentration \leq MDL$$
$$Uncertainty = \sqrt{(Error\ Fraction \times concentration)^2 + (0.5 \times MDL)^2}\ ,$$    (2)
$$when\ concentration > MDL$$

The MDLs used in this study are 0.1 ng/m$^3$, 2 pg/m$^3$, and 2 pg/m$^3$ for GEM, GOM
and PBM, respectively (Tekran Inc., 2010). For RM, the MDL was assumed to be 4
pg/m$^3$. The error fractions were assumed to be 15% of concentrations for Hg forms
and 10% of concentrations for other compounds. This is because most of the
measured GOM and PBM concentrations have low concentrations near or below
MDL as seen in Tables 1-2; thus have large uncertainties as pointed out by Croghan
and Egeghy (2003). Following Polissar et al. (1998), constant uncertainties (100%,
200% and 1000% of the mean/median for GEM, PBM and GOM, respectively) were
used for imputed Hg concentrations, based on the uncertainty distributions of the
below MDL values in the two base cases. This is to down weight the imputed values.
No variables or samples were excluded after input data screening to reflect all
observations. No variables were down-weighted, with the exception of imputed values,
because runs with and without GOM and PBM categorized as "weak" led to similar
results. Other PMF input parameters include: the number of runs was set to 20 to
enable stability evaluation, and the best run was used; the number of the starting seed
was set to 25.
PMF outputs used in this study include source profiles, model performances and
factor contributions. Four factors were retained in each case. The factors were
interpreted based on the comparison of the major variables (>=25%) in each of the
four factors to markers and source profiles in the literature, taking into consideration
NPRI emission sources. Stability indexes of model runs, scaled residual plot,
Obs/Pred scatter plot and Obs/Pred time series were used to evaluate the model
performances for speciated Hg. The impact of each data treatment method on PMF
results was assessed, taking into consideration interpretability of the factors and





model performance of the three Hg forms.

*PCA*

The PCA source apportionment analysis using speciated Hg in 2009 and 2010

was already conducted in another study (Cheng et al., 2013). In this study, different
cases were investigated, as listed in Table 4. Briefly, all compounds were included to
enable comparison with PMF results (Case 2009 and Case 2010), instead of removing
some air pollutants as in Cheng et al. (2013) due to a lack of correlation between those
air pollutants and atmospheric Hg. Pairwise deletion of missing values in Cheng et al.
(2013) was replaced with listwise deletion to be consistent with the PMF model input
which must be a complete data matrix. The PCA runs were conducted using SPSS
22.0 (IBM Corp., USA). Cases 09-C&M and Case 10-C&M were included to evaluate
the effects of weather conditions on factor identification. The components with
eigenvalues greater than 1 were retained for further analysis, following the Kaiser
Criterion (Kaiser, 1960). Principal components after Varimax rotation were interpreted
by comparing the major variables (loadings > 0.25) of the component with the
outcomes of other studies, and by checking NPRI sources in the region (Table S1).

**3.    Results and discussion**
**3.1 PMF - base cases**

In this section, only the two base cases, Case 2009 and Case 2010 are considered.

*PMF sources*

Table 5 and Figures 1-2 present percent concentration of each pollutant

apportioned to each of the four factors. Factor 1 was named Combustion Emission
due to large contributions of $SO_4^{2-}$ (64%) and $HNO_3$ (54%) and a moderate
contribution of GOM (31%). $SO_2$ and $NO_x$ are precursors of $SO_4^{2-}$ and $HNO_3$,
respectively. These precursors are from combustion sources and probably oxidized
during the transport from sources to receptor sites (Liu et al., 2007). The presence of
GOM is consistent with the combustion emission which is one of the GOM sources
(Carpi, 1997). There was little $NH_3$ emissions from point sources near the study site
(Table S1). Thus, the presence of $NH_4^+$ (71%) should be related to the transport and
transformation of $NH_3$ from agriculture emissions as well as other physical and





chemical processes (e.g., aqueous phase chemistry, condensational growth, droplet
evaporation) producing $NH_4^+$ (Zhang et al, 2008; Pitchford et al., 2009). In this
factor, the molar ratio of $NH_4^+$ to $SO_4^{2-}$ is 1.7, although some observed profiles having
ratios greater than 2 (Lee et al, 1999). Ratios less than 2 suggest insufficient amount
of $NH_3$ to neutralize $H_2SO_4$ thus $H_2SO_4$ will react with other compounds to form
sulfate (Pavlovic et al., 2006; Zhang et al., 2008). The moderate contribution of PM
(42%) is consistent with the presence of particulate $SO_4^{2-}$ and $NH_4^+$. Also, $SO_4^{2-}$
accounted for over 50% of PM mass (Table 1). In addition to a lack of major
combustion facilities nearby (Table S1), a strong correlation between $SO_4^{2-}$ and $NH_4^+$
(Tables S2-S3) also suggest formation of secondary aerosols. Therefore, this factor
suggests transported plumes instead of fresh emissions.

Factor 2 was assigned to Industrial Sulfur. The major variables PBM and $SO_2$ are

indicators of coal combustion (Huang et al., 2010). The minor contributions of $HNO_3$
and $NO_3^-$ also suggest combustion sources because their precursor, $NO_x$, is mainly
released by combustion sources (Liu et al., 2007). However, there were no
combustion sources emitting Hg compounds near the KEJ site in 2009 (Table S1).
Therefore, this factor is more likely related to industrial sources in the region.

Factor 3 was named Photochemical Process and Re-emission of Hg due to the

high contributions of ozone (72%), GEM (76%), GOM (69%), PBM (63%), and
moderate contributions of $Ca^{2+}$ (45%) and $K^+$ (37%). The high contribution of ozone
indicates an ozone rich environment, resulting in oxidation of GEM to GOM and the
sequential formation of PBM (Pal and Ariva, 2004; Liu et al., 2007). Although results
of recent studies show that the reaction rate of Hg and ozone has large uncertainties,
the oxidation of Hg by bromine is very fast (Goodsite et al., 2004). The KEJ site is
near the Atlantic, making the oxidation of Hg by bromine applicable. The presence of
$K^+$ is related to soil emission or biomass burning (Andersen et al., 2007), while $Ca^{2+}$
is related to soil/crustal. The site is located in Kejimkujik National Park. Therefore, it
is under the impact of soil emission, emission from the nearby biomass-fired power
station (Table S1), and transported biomass combustion. It was estimated that
re-emission of Hg from biomass burning and land surfaces contributed 13% and 34%
of the global re-emission budget, respectively (Pirrone et al., 2010). Thus, the high
contribution of GEM may be attributable to the re-emission of GEM. The emission





from soil and biomass combustion was also identified in the PCA study at this site
(Cheng et al., 2013).
Factor 4 has high contributions of $Cl^-$ (100%), $Mg^{2+}$ (82%) and $Na^+$ (86%) and
moderate contributions of $Ca^{2+}$ (31%), $K^+$ (39%) and $NO_3^-$ (40%). The presence of
$Na^+$, $Mg^{2+}$, and $Cl^-$ indicates marine aerosols because these elements are rich in sea
water (Huang et al., 1999). The strong correlations among these three compounds
(≥0.89, Tables S2-S3) also suggest a common source. As the sampling site is located
near the Atlantic, the presence of marine aerosols is reasonable. Major production
pathways of $NO_3^-$ include reaction of $HNO_3$ with $NH_3$, sea salt and soil dust
(Pakkanen, 1996). In this factor, the $NO_3^-$ is probably related to the reaction of $HNO_3$
and sea salt. Thus, this factor was named Sea Salt.
As seen in Table 5 and Figures 1-2, the same four factors were identified in year
2009 and 2010. The profiles of each factor were also largely consistent between the
two years. Factor 1 in 2010 is similar to the factor named Combustion Emission in
Case 2009. However, this factor lacks PM (not available in 2010) and has a higher
contribution from $K^+$, which may relate to biomass burning. This factor is assigned to
the same name as in 2009 because the presence of $SO_4^{2-}$ and $HNO_3$ is enough to
identify combustion process (Liu et al., 2007). It should be noted that this factor has a
much smaller constitution of GOM than in 2009. This may be due to a large reduction
in $SO_2$ emissions (2.42 million tons or 32% reduction) from coal-fired power plants
across the United States between 2008 and 2010 (US EPA, 2011). Large reductions in
Hg (-39%) and $SO_2$ (-35%) emissions also occurred in Nova Scotia between 2009 and
2010, as seen in Table S1. However, reduction in Hg emissions is only reflected on
GOM (-75%), while GEM decreased a little and PBM increased slightly.
The major variables of factor 2 are also similar to those of the factor Industrial
Sulfur in Case 2009. However, this factor has a moderate contribution of GOM
instead of PBM in 2009. Factor 3 has similar major variables as the factor named
Photochemistry and Re-emission in Case 2009. Factor 4 is dominated by $Cl^-$ (100%),
$Na^+$ (83%) and $Mg^{2+}$ (75%). This factor was named Sea Salt as in Case 2009.
***PMF source contributions***
The PMF factor contributions of the two base cases are presented in Table S4
(Case 2009) and Table S5 (Case 2010). In both years, factor Photochemistry and





Re-emission had the largest contributions to GEM (averaging 77% and 79% in 2009
and 2010, respectively), GOM (70% and 67%), and PBM (69% and 80%) among all
four factors. In other words, ambient Hg concentrations at the KEJ site were
dominated by photochemistry and re-emission of Hg. Industrial Sulfur had moderate
contributions to GOM (average, 29%) in 2010 instead of PBM in 2009 (21%).
Combustion Emission contributed 25% of GOM in 2009 but 11% each of GEM and
PBM in 2010. The factor Sea Salt only had minor contribution to GEM (14% in 2009
and 9% in 2010) and PBM (<10% in both years). This is not unexpected because
GEM is likely to be oxidized to GOM by the *in situ* photochemical process under the
bromine-rich environment (Obrist et al., 2011). However, this factor has no
contribution to GOM because it was estimated that >80% of GOM in the marine
boundary layer is absorbed by sea salt aerosols and it is sequentially deposited onto
the earth's surface where evasion occurs (Holmes et al., 2009).
*PMF model performance*
Among the three Hg forms, GEM had the best performances in terms of scaled
(i.e. standardized) residual because it had normal distribution and fewer absolute
values of scaled residual greater than 3 in both years (Case 2009 and Case 2010, Table
6). Table 6 also lists the coefficient of determination ($R^2$) and the slope of the
regression line for speciated Hg in Obs/Pred scatter plot (Figures S1-S2), to evaluate
the overall model-measurement agreement. Between the two years, the agreement was
better with GEM in 2010 and PBM in 2009 because of higher $R^2$ values and slope
closer to 1. The low values of $R^2$ and slope in both years indicate the agreement was
poor for GOM.
The Obs/Pred time series of the three Hg forms reveal the model's ability to
reproduce the observational concentrations on a day-to-day basis. In Case 2009, the
Obs/Pred time series (Figure S3) were split into three time periods by the data gaps,
January to February (period 1), March to July (period 2), and October to December
(period 3). GEM had better performances than the other two forms because the peak
values were reproduced by the model in all three periods. However, the modeled
values in period 3 are too low compared to observed concentrations, leading to a
lower $R^2$ (Table 6). The performance for PBM is better than GOM because the
predicted concentrations tracked the observed concentrations well in period 2.





However, PBM concentrations were underestimated and overestimated by the model
in period 1 and period 3, respectively. The GOM concentrations were not reproduced
well with unmatched peak values in period 2, and there was a clear separation of
observed and predicted trend lines in periods 1 and 3, leading to over prediction.
In Case 2010, the time series (Figure S4) were split into two periods,
January-June (period 1) and July-December (period 2), based on a clearly visible
overestimation of GOM concentrations in the second period. The predicted GEM
concentrations tracked the trend of observations well in both periods but with more
fluctuations. The model was unable to reproduce high GOM concentrations in period
1. For PBM, the predicted concentration was rather flat, missing completely the high
concentration episode in spring 2010.
The model-measurement agreement was further quantified with the ratios of
predicted to observed concentrations (Pred/Obs ratio, Figure 3). In both years, the
predicted GEM agreed well with the observed concentrations as supported by the
small range of Pred/Obs ratios (0.56-1.32 in 2009, 0.42-1.43 in 2010) and mean ratios
approaching 1 (0.97 and 0.98). On an annual basis, the observed GEM concentrations
were also well reproduced because the ratios of predicted to observed annual means
(annual Predmean/Obsmean) were almost 1 (0.97 and 0.98) (Tables S4-S5).
Compared with GOM, PBM had better agreement between the predicted and observed
concentrations with a smaller range of Pred/Obs ratios (0.40-13.4 and 0.14-18.3 vs.
0.13-53 and 0-193) and mean ratios closer to 1 (2.09 and 1.98 vs. 5.89 and 4.44). In
spite of large sample to sample variability in the Pred/Obs ratios, the model
performance was very good for PBM (annual Predmean/Obsmean ratio of 1.03 and 1)
and reasonable for GOM (0.86 and 1.34) in reproducing annual means.
***Comparison between PMF in year 2009 and 2010***
Overall, the interpretability of the factors was similar in the two years. The same
factors were observed in 2009 and 2010, and most factor contributions were highly
consistent between the two years. Among the three Hg forms, PMF reproduced GEM
concentrations well in both years. Possible reasons of poor performance on PBM and
GOM include lower concentration levels and much higher percentages of readings
below MDL (Tables 1-2) leading to large uncertainties. However, the differences in



sample size (161 in 2009 vs. 290 in 2010) and fractions of below MDL values (Tables
1-2) alone could not explain the mixed results of poor performance on GOM in 2009
and PBM in 2010. Further examination of time series (Figures S3 and S4) suggests
that the reduced performance could also be attributable to high concentration episodes
in GOM in 2009 and PBM in 2010. The impact of Hg data treatment on PMF results
was investigated and the results are presented in section 3.4.

**355  3.2 PCA components**

*Case 09-C*
The component loadings of Case 09-C are presented in Table 7. PC1 was named
Combustion/industrial Emission due to positive loadings of PBM, PM, $O_3$, $SO_2$,
$HNO_3$, $Ca^{2+}$, $K^+$, $NO_3^-$, $NH_4^+$, and $SO_4^{2-}$. Most major compounds except $O_3$ were also
found in a component named "transport of combustion and industrial emissions" in
another PCA study using the same dataset (Cheng et al., 2013). The high loadings of
secondary pollutants $HNO_3$, $NO_3^-$, and $SO_4^{2-}$ indicate the factor represents transport of
combustion/industrial emission because their precursors ($NO_x$ and $SO_2$) are mainly
emitted by combustion/industrial sources (Liu et al., 2007). The precursors may be
oxidized during the transport process. The moderate loading of $O_3$ is also related to
the transport of combustion emission because the precursors of $O_3$ ($NO_x$ and VOC)
are emitted from mobile and stationary combustion sources. Ammonia is likely related
to the transport of agriculture emissions and reaction of $NH_3$ and $H_2SO_4$ or $HNO_3$
(Pichford et al., 2009).
PC2 has high loadings of $Na^+$, $Mg^{2+}$, $Cl^-$, and $K^+$ and and moderate loadings of
$Ca^{2+}$. Those compounds indicate marine aerosols (Huang et al., 1999). The moderate
loading of $NO_3^-$ is likely due to the reaction of $HNO_3$ and sea salt (Pakkanen, 1996).
As in the PMF factor interpretation, the identification of component Sea Salt is
relevant because the monitoring site is near the Atlantic.
PC3 has positive loadings of GEM, GOM, PBM and $O_3$. The positive loadings on
$O_3$ and GOM indicate the photochemical production of GOM (Huang et al., 2010).
The positive loading of GEM is somewhat unexpected because the photochemical
production of GOM consumes GEM thus leading to opposite signs of GEM and GOM
(e.g. Huang et al., 2010). However, daily average concentrations were used in this





study instead of two-hour means in Huang et al. (2010). The daily GEM and GOM
were indeed positively correlated (r=0.37 in 2009, Table S2; 0.31 in 2010, Table S3).
Using the same dataset, Cheng at al. (2013) conducted further analysis on $O_3$
concentrations and %GOM/TGM (TGM=GEM+GOM) ratios. The ratio is indicative
of the degree of oxidation. The results showed that the %GOM/TGM ratio increased
with $O_3$ when $O_3$ concentrations were greater than 40 ppb, suggesting gas phase
oxidation of Hg at this coastal site. Therefore, this factor was named Photochemical
Production of GOM.
PC4 represents Gas-particle Partitioning of Hg. The negative loading of PBM and
the positive loading of GOM indicate the partition process. The positive loadings of
$Ca^{2+}$ and $K^+$ suggest soil aerosols (Cheng et al., 2012) which could be abundant at the
Kejimkujik National Park.
Three out of four components (Combustion/industrial Emission, Photochemical
Production of GOM and Gas-particle Partitioning of Hg) have significant association
with ambient Hg concentrations at the site, while Sea Salt has little.
*Case 09-C&M*
Five principal components are extracted when meteorological data were included
in PCA (Case 09-C&M, Table 7). The loadings in PC1-PC4 are similar with the
loadings of PC1, PC2, PC4, PC3 in Case 09-C, respectively. Thus the names of those
four components were retained. The inclusion of meteorological parameters resulted
in small loadings of relative humidity (-0.26) in PC1 and wind speed (0.32) in PC2, as
well as a moderate loading of wind speed (0.52) in PC4. A large loading of
temperature (0.94) was observed in PC3. The opposite signs of temperature and PBM
are consistent with the gas-particle partitioning process because low temperatures
favor the formation of PBM (Rutter and Schauer, 2007). The lack of GEM in PC3
(Case 09-C&M) did not affect the identification of this factor, because the partitioning
of GEM onto particles is much weaker than that of GOM (Liu et al., 2007).
PC5 was derived mostly from meteorological variables. The negative loading of
GOM and positive loadings of relative humidity and precipitation suggest removal of
GOM by dew, cloud and precipitation (Cheng et al., 2013). The loading of GOM is
small, nonetheless consistent with the wet deposition process because GOM is more
easily removed compared to GEM due to its higher water solubility (Gaffney and



Marley, 2014). Therefore, this component was named Hg Wet Deposition.
Similar to Case 09-C, all components except Sea Salt are associated with ambient
Hg concentrations. After the inclusion of meteorological data, each factor contains at
least one meteorological parameter. The presence of meteorological variables did not
contribute to the determination of the components except a new component Hg wet
deposition was identified.
*Case 10-C*
The component loadings of Case 10-C are listed in Table 8. PC1 was named
Combustion Emission. The positive loadings of $HNO_3$, $NO_3^-$ and $SO_4^{2-}$ are indicative
of transport of combustion emission because their precursors ($NO_2$ and $SO_2$) are
mainly released by combustion emissions (Liu et al., 2007). The high positive loading
of $NH_4^+$ represents transport of agriculture emissions of ammonia which may react
with $H_2SO_4$ and $HNO_3$ during the transport process (Pitchford et al., 2009). The
positive loadings of $Ca^{2+}$ and $K^+$ indicates biomass burning from wildfires or
biomass-fueled power station (Andersen et al., 2007).
PC2 was named Sea Salt due to high loadings of $Na^+$, $Mg^{2+}$, and $Cl^-$, because
these three compounds are rich in sea water (Huang et al., 1999). PC3 has the same
major variables as the component Photochemical Production of GOM in 2009.
Therefore, PC3 was also named as such.
PC4 was assigned to Industrial Source. The positive loadings of GOM and $SO_2$
indicate coal combustion (Lynam and Keeler, 2006), although no combustion facilities
were reported near the KEJ site in 2010 (Table S1). The positive loadings of $SO_4^{2-}$ and
$HNO_3$ are consistent with the transport of industrial emissions which release their
precursors, $SO_2$ and NOx (Liu et al., 2007). Therefore, this factor was named
Industrial Source. Two out of four factors (i.e. Photochemical Production of GOM and
Industrial source) have significant association with Hg compounds.
*Case 10-C&M*
As shown in Table 8, five principal components are extracted in Case 10-C&M.
The loadings in PC1-PC3 and PC5 are similar with the loadings of PC1-PC4 in Case
10-C, respectively. Thus the name of those four components were retained. PC4 in
Case 10-C&M was named Hg Wet Deposition due to negative loadings of GOM and
PBM and positive loadings of relative humidity, wind speed and precipitation, similar



with PC5 in Case 09-C&M (Table 7). Three out of five components (i.e.
Photochemical Production of GOM, Industrial Source, and Hg Wet Deposition) were
associated with Hg concentrations. The influence of meteorological data on
identification of components were also similar to in 2009.
***Comparison between PCA in year 2009 and 2010***
In each year, four components were extracted in PCA with air pollutants only.
The two common factors between the two years are Photochemical Production of
GOM and Sea Salt. The former has a strong association with Hg compounds, while
the latter has little. Component Gas-particle Partitioning of Hg was only identified in
2009, likely due to a lower percentage of PBM readings <MDL than those in 2010
(Table 9, Case 2009 and 2010). It is also consistent with strong correlations between
temperature as well as GOM and PBM (r=0.46 and -0.43, Table S2) in 2009 but
non-significant or weak correlations (r=-0.04, and -0.16, Table S3) in 2010.
The component Combustion/industrial Emission in 2009 affected PBM and $SO_2$
levels. It was split into two components in 2010, Combustion Emission and Industrial
Source. The former was no longer strongly associated with any of the three Hg forms,
while the latter was associated with GOM and $SO_2$. This is probably due to the
reduction of coal combustion in Canada and the USA, evident by lower provincial Hg
(reduction of 39%) and $SO_2$ emissions (-35%) in 2010 (Table S1). The reductions in
GEM, GOM, and $SO_2$ concentrations at the KEJ site were 3%, 75%, and 43%
respectively in 2010 (Tables 1-2). The shifting of PBM & $SO_2$ relation in 2009 to
GOM & $SO_2$ in 2010 is sustained by a strong correlation between PBM and $SO_2$
(r=0.63, Table S2) in 2009, but little correlation (r=0.06) accompanied by a moderate
correlation between GOM and $SO_2$ (r=0.30) (Table S3) in 2010. The shift is also
consistent with the PMF results where Industrial Sulfur accounted for 21% of PBM in
2009 (Table S4) but 29% of GOM in 2010 (Table S5).
In both years, inclusion of meteorological parameters did not affect the
identification of the four factors from air concentrations. However, relative humidity
and precipitation yielded an additional component named Hg Wet Deposition.
Overall, the PCA results were largely consistent between the two years, in terms
of the number of components, impact of meteorological parameters, and major
processes associated with ambient Hg. The changing emissions/concentrations and the



resultant correlations among monitored air pollutants from one year to another are
reflected in the limited shifting of variable loadings.            .

**3.3 Comparison of PMF and PCA results**

The PCA loadings and the factor profiles as well as factor contributions in PMF

model have very different meanings. In PCA, variables with large loading indicate
their correlation or association with that component derived from all samples. In PMF,
presence of variables in profiles indicates their contribution to that source/process
derived from all samples, while the contribution values are further quantified in
source contribution tables of each sample. Therefore, a direct comparison between the
PMF and PCA results is not feasible. However, the similarities and differences in the
major sources/processes identified by each approach, chemical markers in each factor
profile or component, and the impact/association of factors/components on Hg could
reveal strength and weakness of each method.

A comparison of Table 5 and Tables 7-8 (cases with air concentrations only)

shows that $Na^+$, $Cl^-$, and $Mg^{2+}$ are markers of Sea Salt in both PMF and PCA.
Similarly, GEM, GOM, PBM and $O_3$ indicate Photochemistry. Both methods suggest
strong contribution to or association between Hg compounds and photochemistry, but
weak with Sea Salt. Both methods identified combustion and industrial sources, while
the variables in factors/components differed to some extent. Furthermore, combustion
and industrial were separate sources in PMF in both years and in PCA in 2010, but
combined as one component in PCA in 2009. Overall, PMF profiles are more
consistent between the two years, while the PCA loadings are more sensitive to
correlation among variables. However, the shift of PBM & $SO_2$ to GOM & $SO_2$
loadings in PCA between the two years is consistent with the shift of those two pairs
in Combustion & Industrial Sulfur profiles/contributions in PMF. On the other hand,
Gas-particle Partitioning of Hg was only recognized in PCA. This is because the
identification of this factor relies on negative association between PBM and GOM
(Table 7), but such association is not reflected in PMF due to its non-negative nature.
This is one of the limitations of PMF. Furthermore, the inclusion of meteorological
conditions in PCA enables identification of a new component related with weather
conditions. The good agreement between PMF and PCA outputs is consistent with a




comparison of receptor models in PM source appointment (Viana et al., 2007). A
common weakness of PCA and PMF is the suggestiveness of factors/components.
Other techniques, such as back trajectories, have been used in previous studies to
verify some factors (Cheng et al. 2015). Overall, when accompanied by model
performance evaluation, PMF results are with more confidence.

**3.4 Sensitivity of PMF results to data treatment**
**3.4.1 Year 2009**
*Case 09+mean & Case 09+median*

The factor profiles of the six PMF cases in 2009 are displayed in Figure 1. In

Case 09+mean and Case 09+median, all four factors have similar profiles as in Case
2009. Compare with the base case, factor 3 (Photochemistry and Re-emission of Hg)
has a higher contribution by $NO_3^-$, however it is common to observe $NO_3^-$ from soil
emissions (Parmar et al., 2001). GOM has a much smaller contribution in factor 1
(Combustion Emission) (Figure 1, Table S4). This is likely because the correlation
coefficients between GOM, $NH_4^+$ and $SO_4^{2-}$ become insignificant after imputation
(Table S6). Consequently, GOM is not strongly related to that factor which is
dominated by $NH_4^+$ and $SO_4^{2-}$. Changing correlation among variables is a
shortcoming of imputation (Huang et al., 1999).

*Case 09+RM & Case 09-RM*

As shown in Figure 1 and Table S4, by combining GOM and PBM into RM,

RM replaced PBM instead of GOM in related factors as major variables with similar
contributions. This is because the median concentration of PBM is approximately 5
time of the median concentration of GOM (Table 9). Once these two forms are
combined to RM, the variance of RM is dominated by PBM. The presence of other
compounds including GEM in factor profiles/contributions in these two cases are
similar to those in Case 2009.
*Case 09ScaleRM*

The factor profiles were similar to those in Case 2009 (Figure 1). The same can

be said about factor contributions to speciated Hg (Table S4).





*Performance*
Case 09-RM, Case 09+RM and Case 09ScaleRM have similar performances with
Case 2009, on distribution of scaled residuals (Table 6). Imputation (Case 09+mean
and Case 09-median) worsened the performance because the scaled residuals are
concentrated near zero for gaseous Hg.
In terms of the coefficients of determination ($R^2$) and the slopes of the regression
line for speciated Hg in Obs/Pred scatter plot (Table 6, Figure S1), imputation (Case
09+mean and Case 09+median) deteriorated PMF performance compared to the base
case. This is not unexpected because the use of a constant imputation value reduced
the variance in observed concentrations (Table 9). The similar performances on GEM
in Case 2009, Case 09+RM, Case 09-RM, and Case 09ScaleRM indicate combining,
excluding, or scaling GOM and PBM, respectively, did not affect the performance on
GEM. The performances on RM are similar to that of PBM in Case 2009 because the
RM concentrations are dominated by PBM. Using scaling factors to increase GOM
and PBM concentrations resulted in better performances on those two forms than in
the base case. This is attributable to a significant reduction in percent of
concentrations below MDL (Table 9).
The changes in model performance are more evident in the observed and
predicted time series (Figure S3). Compared with the base case, imputation led to
more fluctuation in the predicted GEM values, thus slightly worse. RM had better
model-measurement agreement than GOM or PBM as individual compound. The
agreement was also improved by scaling GOM or PBM. The peak values (PBM in
period 1 and both forms in period 2) were better reproduced and the over prediction in
period 3 with low concentrations was greatly corrected.
Compared with the base case, the distributions of the ratios of predicted to
observed Hg concentrations and the ratio of predicted to observe annual means
changed little for GEM among the six cases (Figure 3, Table S4). Scaling GOM and
PBM improved model-measurement agreement of those two forms, evident by a
much narrower range and a shift toward smaller values in the distribution of ratios.
**3.4.2 Year 2010**





### *Case 10+mean & Case 10+median*


Factor profiles (Figure 2) and contributions (Table S5) after imputation have
minor changes compared to those in Case 2010. However, less changes were observed
with the use of median imputation. The small deviations after imputations is probably
because only a small fraction (4%) of Hg concentrations were missing in 2010 than in
2009 (31-41%). Although $HNO_3$, $SO_2$, and inorganic ions have up to 19% missing
values (Table 2), the correlations between each of the three Hg forms and other
compounds changed little (Table S7).

### *Case 10+RM & Case 10-RM*


The impact of combining or removing GOM and PBM (Figure 2, Table S5) is the
same as in 2009. The dominance of PBM in RM is stronger in 2010 with the ratio of
median PBM to median GOM concentration being approximately 10 (Table 9).
Overall, excluding or combining GOM and PBM did not affect the source
identification in PMF model in both years (Figures 1 and 2). However, the
identification of the factors relying on GOM or PBM only (e.g. gas-particle
partitioning of Hg) may be affected after combining or excluding GOM and PBM.
In this study, such factors were not encountered in PMF. Nonetheless, excluding or
combining GOM and PBM did affect the source contributions. After combining GOM
and PBM, factors contributing to GOM only (Combustion Emission, 2009; Industrial
Sulfur, 2010, Table 10) did not contribute to any Hg forms, and the factor contributing
to PBM only (Industrial Sulfur, 2009) was contributing to RM, due to dominance of
PBM in RM. In both years, using three Hg forms instead of GEM only led to more Hg
sources/processes identified. Therefore, monitoring speciated Hg could help us better
understand Hg cycling.

### *Case 10ScaleRM*


The factor profiles and contributions of Case 10ScaleRM are similar to those in
Case 2010 (Figure 2, Table S5). A noticeable deviation is the much smaller
contribution by GOM in factor 2 compared to Case 2010. However, factor 2 was still
assigned to Industrial Sulfur because of the presence of $SO_2$ and $NO_3^-$.

### *Performance*




Firstly, the distribution of scaled residuals as well as $R^2$ value and the slope of the
regression line for speciated Hg in Obs/Pred scatter plot was evaluated for the six
cases (Table 6, Figure S2). Similar to 2009, the comparable performances observed in
Case 2010, Case 10-RM, Case 10+RM, and Case 10ScaleRM indicate that the model
performance on GEM is insensitive to excluding, scaling, or combining GOM and
PBM to RM. Case 10ScaleRM also has the best performances on GOM and PBM
among all the cases in 2010. Unlike in 2009, the negative impact of imputation was
smaller when median value was used, compared with the mean imputation.
Secondly, in the observed and predicted time series (Figure S4), imputation
resulted in more severe fluctuation in predicted GEM concentration as in 2009, but
less so when median values were used. Scaling of GOM or PBM also improved the
reproducibility of day-to-day variability in the observed values, owing to a large
reduction in concentrations below MDL (Table 9). Among the 6 cases, the most
significant change is in PBM with imputation. There were additional high
concentration episodes in early 2010 when imputation of non-Hg compounds brought
back Hg concentrations otherwise removed by listwise deletion in the base case,
leading to increased standard deviation (Table 9). Those peaks were completely
missed by the model, leading to deteriorated agreement.
Finally, the distributions of the ratios of predicted to observe Hg concentrations
and the ratio of predicted to observe annual means changed little among the first five
cases in 2010 (Figure 3 and Table S5). The exceptions are under prediction of the
annual mean of PBM in the two imputation cases and over prediction for RM.
Compared with the base case, the distribution of ratios for GOM and PBM became
narrower and shifted toward smaller values, but leading to under prediction of PBM.
**3.4.3 Comparison of 2009 and 2010 among different data treatments**
The different characteristics of Hg forms led to different impact of data treatment
on model results and performances in the two years.  Imputation using geometric
mean and median values led to minor changes in factor profiles in both years, with
more variations in contributions of Hg forms in 2009 but non-mercury compounds in
2010. This is likely because the Hg and non-Hg compounds were missing at a larger
percentage in 2009 and 2010, respectively. The lack of significant impact is likely due



to already high sample to compound ratios (161 samples/15 compounds in 2009, 290
samples/14 compounds in 2010, Tables 1-3). Huang et al. (1999) have reported that
mean imputation generally yielded better PMF results than listwise deletion,
especially with higher percentage of missing values. Particularly, composition of
crustal and marine factors were closer to those of crust and sea water. Imputation
resulted in degraded performance on all three Hg forms, but for different reasons. For
GEM, it is largely due to more fluctuation than the already over predicted one in the
base case in both years. For PBM in 2010, the peak values otherwise removed in
listwise deletion (base case) are beyond the model's ability to match. This seems to be
a random occurrence and is an uncertainty of imputation. Between geometric mean
and median imputations, the impact was similar in both years for each of the three Hg
forms. The exception is with median imputation in 2010, there was less deviation in
factor profile/contribution from the base case. The reason is unclear because the
difference in geometric mean and median was very small for GEM in both years and
slightly greater in 2009 for GOM and PBM (Tables 1-2).

In both years, some changes in the factor profiles and factor contributions but

little changes in model performances were observed in the cases excluding GOM and
PBM. Scaling GOM and PBM or combining them into RM improved
model-measurement agreement, suggesting the approach is effective in both year in
spite of large percentages of below MDL values (GOM, 78% in 2009 vs. 96% in 2010;
PBM, 48% in 2009 vs. 46% in 2010, Tables 1-2). The improvement is largely
attributable to reduction in concentrations below MDL (Table 9) which in turn
reduced PMF uncertainty expressed in equation (2). Another benefit of using a
variable scaling factor is reduced data variability as indicated by smaller coefficients
of variation in Table 9. PMF is more likely reproducing well compounds with less
variability. However, there is little evidence that the scientific uncertainties of scaled
GOM and PBM concentrations are indeed reduced from that of the original dataset.



## 4. Conclusions

Source apportionment analysis was conducted with PMF and PCA using concentrations of speciated Hg and other air pollutants collected at KEJ site in 2009 and 2010. Year 2010 was characterized by reduced Hg and $SO_2$ emissions compared with 2009. However, GOM is more sensitive to the decrease in Hg emissions while GEM and PBM are not, underscoring the benefits of speciated Hg measurements. It was found that consideration of emission inventories and correlation among air pollutants is useful in factor/component interpretation.

Using PMF, the nature of each of the four factors identified was the same in 2009 and 2010. In both years, ambient concentration of all three Hg forms at the KEJ site were dominated by contributions from factor Photochemistry and Re-emission, and the contribution by Sea Salt was the smallest. However, slight variations between the two years were observed in the contributions by the other two factors (Combustion Emission, Industrial Sulfur).

Good agreement was found between PMF and PCA results. In each year, four components were extracted in PCA with air pollutants only. Three or four of them overlapped with factors obtained in PMF. PCA results suggest little association between Hg and Sea Salt, consistent with PMF. Furthermore, PMF and PCA had similar shift of source profile/contribution from one year to another, suggesting both methods were able to respond to changing concentration levels, and interrelationships among the air pollutants. In both years, inclusion of meteorological parameters in PCA led to extraction of an additional component Hg Wet Deposition while the identification of other components was not affected. Therefore, PCA is superb to PMF in terms of identifying factors related to atmospheric processes. With regards to atmospheric processes represented by negative correlation among variables, such as Gas-particle Partitioning of Hg (Table 8), PCA is more likely to identify them because component loadings reflect correlations, while it is difficult for PMF because its variable contributions in source profile are all positive.

A comprehensive PMF model performance evaluation was conducted for each of the three Hg forms. Between the two years, the model performance was comparable. In both years, the observed daily GEM concentrations were well reproduced, but



relatively poor for GOM and PBM. On an annual basis, the model-measurement agreement of annual mean concentrations were excellent for GEM, very good for PBM and acceptable for GOM.

The sensitivity of PMF results and model performance to different approaches of dealing with missing values and concentrations with large uncertainties was investigated. In our study of more than 160 samples with 15 or 14 air pollutants, increasing the sample size by geometric mean or median imputation of missing values is not effective in improving the model performance. With over 70% GOM and over 40% PBM concentrations below MDL in our dataset, the impact of large measurement uncertainties in GOM and PBM is much more significant. Scaling GOM and PBM to increase their concentrations or combining them to reactive mercury is effective in improving the model-measurement agreement. The identification of sources/processes and their contributions to speciated Hg are relatively insensitive to any of the data treatment options considered. The exception is that less sources/processes affecting ambient Hg were identified when GOM and PBM were excluded, further underlining the importance of monitoring speciated Hg.

The good agreement between PCA and PMF results in both years is encouraging although these two methods bear little resemblance. PMF partitions observed concentrations by solving mass balance equations, while PCA is a data reduction tool to explain majority of variances in the entire dataset with a small number of components. Our observation was made possible by the use of multiple-year dataset. Future studies should conduct more PMF and PCA comparisons to validate our findings.

Overall, PMF results are quantitative and with more confidence with model performance evaluation. However, when ancillary air pollutant data are available, it is recommended to carry out both PCA and PMF simulations to verify the sources/processes identified.

Our PMF results suggest that PMF has difficulties reproducing daily concentrations of GOM and PBM, because of high concentration episodes and large uncertainties due to low concentrations and large percentage of below MDL values. More attention should be devoted to those issues in future studies.



**Acknowledgements:** Funding of this project was provided by Environment Canada
and National Sciences and Engineering Research Council of Canada. The authors
acknowledge John Dalziel and Rob Tordon of Environment Canada for providing
mercury data and US EPA for the PMF model used in this study.

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




**List of Tables**





**List of Figures**





Table 1. General statistics of air pollutant concentrations (in µg/m$^3$ unless otherwise noted) in 2009.

| Compound | Percent of missing values | Method detection limit (MDL) | Percent of values <MDL | Geometric Mean | Median | Mean | Standard deviation | Coefficient of variability (%) |
|---|---|---|---|---|---|---|---|---|
| GEM (ng/m$^3$) | 31% | 0.1 | 0% | 1.37 | 1.41 | 1.39 | 0.26 | 18.7 |
| GOM (pg/m$^3$) | 32% | 2 | 78% | 0.57 | 0.42 | 1.77 | 3.70 | 209 |
| PBM (pg/m$^3$) | 41% | 2 | 48% | 1.78 | 2.15 | 2.81 | 2.72 | 96.8 |
| PM | 20% | 1 | 9% | 2.71 | 2.91 | 3.44 | 2.49 | 72.4 |
| O$_3$ | 0% | 4.3 | 0% | 59.4 | 62.1 | 62.4 | 19.1 | 30.6 |
| SO$_2$ | 3% | 0.002 | 0% | 0.20 | 0.22 | 0.40 | 0.51 | 128 |
| HNO$_3$ | 3% | 0.05 | 12% | 0.13 | 0.12 | 0.19 | 0.22 | 116 |
| Ca$^{2+}$ | 1% | 0.002 | 0% | 0.05 | 0.05 | 0.06 | 0.04 | 66.7 |
| K$^+$ | 1% | 0.029 | 17% | 0.04 | 0.03 | 0.04 | 0.03 | 75.0 |
| Na$^+$ | 1% | 0.05 | 9% | 0.25 | 0.30 | 0.43 | 0.47 | 109 |
| Mg$^{2+}$ | 1% | 0.0004 | 2% | 0.04 | 0.04 | 0.06 | 0.06 | 100 |
| Cl$^-$ | 1% | 0.046 | 23% | 0.19 | 0.23 | 0.46 | 0.64 | 139 |
| NO$_3^-$ | 1% | 0.06 | 9% | 0.18 | 0.17 | 0.28 | 0.39 | 139 |
| NH$_4^+$ | 1% | 0.001 | 0% | 0.19 | 0.17 | 0.28 | 0.32 | 114 |
| SO$_4^{2-}$ | 1% | 0.05 | 0% | 0.78 | 0.76 | 1.14 | 1.27 | 111 |





Table 2. General statistics of air pollutant concentrations (in µg/m$^3$ unless otherwise noted) in 2010, MDL same as in Table 1.

| Compound | Percent of missing values | Percent of values <MDL | Geometric Mean | Median | Mean | Standard deviation | Coefficient of variability (%) |
|---|---|---|---|---|---|---|---|
| GEM (ng/m$^3$) | 4% | 0% | 1.34 | 1.38 | 1.35 | 0.17 | 12.6 |
| GOM (pg/m$^3$) | 4% | 96% | 0.27 | 0.21 | 0.44 | 0.64 | 145 |
| PBM (pg/m$^3$) | 4% | 46% | 2.08 | 2.20 | 3.40 | 4.13 | 121 |
| O$_3$ | 1% | 0% | 62.2 | 63.4 | 64.5 | 16.6 | 25.7 |
| SO$_2$ | 19% | 1% | 0.10 | 0.13 | 0.23 | 0.31 | 135 |
| HNO$_3$ | 19% | 25% | 0.10 | 0.10 | 0.18 | 0.22 | 122 |
| Ca$^{2+}$ | 19% | 0% | 0.04 | 0.04 | 0.07 | 0.13 | 186 |
| K$^+$ | 19% | 46% | 0.04 | 0.03 | 0.06 | 0.07 | 117 |
| Na$^+$ | 19% | 16% | 0.20 | 0.24 | 0.40 | 0.53 | 133 |
| Mg$^{2+}$ | 19% | 0 % | 0.03 | 0.04 | 0.05 | 0.06 | 120 |
| Cl$^-$ | 19% | 27% | 0.14 | 0.15 | 0.46 | 0.83 | 180 |
| NO$_3^-$ | 19% | 21% | 0.14 | 0.13 | 0.25 | 0.36 | 144 |
| NH$_4^+$ | 19% | 0% | 0.16 | 0.15 | 0.30 | 0.57 | 190 |
| SO$_4^{2-}$ | 19% | 0% | 0.69 | 0.64 | 1.11 | 1.65 | 149 |





Table 3. PMF case design with different treatments of speciated Hg data.

| Case | | Input variables | Treatment of missing value | Sample size | |
|---|---|---|---|---|---|
| 2009 | 2010 | | | 2009 | 2010 |
| 2009 (base case) | 2010 (base case) | All compounds | Excluding listwise | 161 | 290 |
| 09+Mean | 10+Mean | All compounds | Geometric mean imputation | 365 | 365 |
| 09+Median | 10+Median | All compounds | Median imputation | 365 | 365 |
| 09+RM | 10+RM | All compounds, but combining GOM & PBM to RM | Excluding listwise | 161 | 290 |
| 09-RM | 10-RM | All compounds, except GOM & PBM | Excluding listwise | 201 | 290 |
| 09ScaleRM | 10ScaleRM | All compounds, GOM & PBM scaled | Excluding listwise | 161 | 290 |





Table 4. PCA input and set-up.

| Case | Year | Input variables | Sample size | Other settings |
|---|---|---|---|---|
| 09-C | 2009 | All compounds | 161 | 1) Missing value: Listwise deletion |
| 09-C&M | 2009 | All compounds and meteorological parameters | 159 | 2) Components to keep: eigenvalues >1) |
| 10-C | 2010 | All compounds | 290 | 3) Rotation: Varimax |
| 10-C&M | 2010 | All compounds and meteorological parameters | 285 | 4) Cut-off value for major loadings: 0.25 |





Table 5. Factor profiles (concentration >25%, between 20% and 25% in parenthesis) of Case 2009 and Case 2010.

| Compound | 2009 F1 | F2 | F3 | F4 | 2010 F1 | F2 | F3 | F4 |
|---|---|---|---|---|---|---|---|---|
| GEM | | | 76 | | | | 79 | |
| GOM | 31 | | 69 | | | 37 | 59 | |
| PBM | | 29 | 63 | | | | 81 | |
| PM | 42 | | 34 | | - | - | - | - |
| O$_3$ | | | 72 | | | | 80 | |
| SO$_2$ | | 82 | | | | 93 | | |
| HNO$_3$ | 54 | (21) | (25) | | 64 | 26 | | |
| Ca$^{2+}$ | (19) | | 45 | 31 | | 29 | 36 | (21) |
| K$^+$ | (22) | | 37 | 39 | 51 | | 27 | (23) |
| Na$^+$ | | | | 86 | | | | 83 |
| Mg$^{2+}$ | | | | 83 | | | | 75 |
| Cl$^-$ | | | | 100 | | | | 100 |
| NO$_3$ | (25) | (23) | | 40 | | 41 | (23) | |
| NH$_4^+$ | 71 | | | | 87 | | | |
| SO$_4^{2-}$ | 64 | | | | 79 | | | |
| Factor | Combustion emission | Industrial sulfur | Photochemistry & re-emission of Hg | Sea salt | Combustion emission | Industrial sulfur | Photochemistry & re-emission of Hg | Sea salt |





Table 6. PMF model performances on speciated mercury in 2009 and 2010.

| Hg form | Case | Distribution | Number of scaled residuals greater than 3 | Coefficient of determination ($R^2$) | Slope of regression line |
|---|---|---|---|---|---|
| GEM | 09 | Normal | 0 | 0.28 | 0.59 |
| | 09+mean | Concentrated near zero | 5 | 0.17 | 0.57 |
| | 09+median | Concentrated near zero | 5 | 0.15 | 0.54 |
| | 09+RM | Normal | 0 | 0.29 | 0.59 |
| | 09-RM | Normal | 1 | 0.25 | 0.59 |
| | 09ScaleRM | Normal | 0 | 0.28 | 0.58 |
| | 10 | Normal | 2 | 0.46 | 1.29 |
| | 10+mean | Normal | 19 | 0.32 | 1.26 |
| | 10+median | Normal | 2 | 0.41 | 1.26 |
| | 10+RM | Normal | 2 | 0.46 | 1.31 |
| | 10-RM | Normal | 2 | 0.47 | 1.31 |
| | 10ScaleRM | Normal | 1 | 0.44 | 1.19 |
| GOM | 09 | Right skewed | 17 | 0.23 | 0.09 |
| | 09+mean | Concentrated near zero, right skewed | 17 | 0.08 | 0.05 |
| | 09+median | Concentrated near zero, right skewed | 19 | 0.09 | 0.05 |
| | 09+RM | - | - | - | - |
| | 09-RM | - | - | - | - |
| | 09ScaleRM | Right skewed | 26 | 0.33 | 0.18 |
| | 10 | Narrower | 0 | 0.31 | 0.29 |
| | 10+mean | Narrower | 0 | 0.23 | 0.22 |
| | 10+median | Narrower | 0 | 0.28 | 0.28 |
| | 10+RM | - | - | - | - |
| | 10-RM | - | - | - | - |
| | 10ScaleRM | Narrower | 0 | 0.42 | 0.33 |
| PBM | 09 | Normal | 5 | 0.57 | 0.39 |
| | 09+mean | Right skewed | 6 | 0.33 | 0.32 |
| | 09+median | Right skewed | 6 | 0.34 | 0.34 |
| | 09+RM | Right skewed (RM) | 8 (RM) | 0.48(RM) | 0.31(RM) |
| | 09-RM | - | - | - | - |
| | 09ScaleRM | Left skewed | 2 | 0.59 | 0.48 |
| | 10 | Right skewed | 14 | 0.13 | 0.09 |
| | 10+mean | Right skewed | 28 | 0.15 | 0.09 |
| | 10+median | Right skewed | 29 | 0.16 | 0.08 |
| | 10+RM | Right skewed (RM) | 5 | 0.19 | 0.15 |
| | 10-RM | - | - | - | - |
| | 10ScaleRM | Normal | 18 | 0.25 | 0.24 |





Table 7. PCA component loadings (>0.25) of Case 09-C and Case 09-C&M.

| Variable | Case 10-C PC1 | PC2 | PC3 | PC4 | Case 10-C&M PC1 | PC2 | PC3 | PC4 | PC5 |
|---|---|---|---|---|---|---|---|---|---|
| GEM | | | 0.86 | 0.27 | | | | 0.80 | |
| GOM | | | 0.26 | 0.84 | | | 0.64 | 0.41 | -0.29 |
| PBM | 0.63 | | 0.50 | -0.33 | 0.59 | | -0.47 | 0.34 | |
| PM | 0.80 | | | | 0.81 | | | | |
| O$_3$ | 0.50 | | 0.70 | | 0.47 | | | 0.72 | -0.27 |
| SO$_2$ | 0.88 | | | | 0.86 | | | | |
| HNO$_3$ | 0.86 | | | 0.34 | 0.88 | | | | |
| Ca$^{2+}$ | 0.59 | 0.39 | | 0.45 | 0.60 | 0.38 | 0.33 | | |
| K$^+$ | 0.29 | 0.70 | | 0.33 | 0.36 | 0.66 | 0.39 | | |
| Na$^+$ | | 0.97 | | | | 0.96 | | | |
| Mg$^{2+}$ | | 0.95 | | | 0.28 | 0.95 | | | |
| Cl$^-$ | | 0.97 | | | | 0.98 | | | |
| NO$_3^-$ | 0.73 | 0.48 | | | 0.76 | 0.45 | | | |
| NH$_4^+$ | 0.92 | | | | 0.94 | | | | |
| SO$_4^{2-}$ | 0.86 | | | | 0.88 | | | | |
| Temperature | - | - | - | - | | | 0.94 | | |
| Relative humidity | - | - | - | - | -0.26 | | | | 0.79 |
| Wind speed | - | - | - | - | | 0.32 | | 0.52 | 0.49 |
| Precipitation | - | - | - | - | | | | | 0.79 |
| Component | Combustion/industrial emission | Sea salt | Photochemical production of GOM | Gas-particle partition of Hg | Combustion/industrial emission | Sea salt | Gas-particle partition of Hg | Photochemical production of GOM | Hg wet deposition |
| Variance explained | 37% | 25% | 11% | 9% | 30% | 20% | 10% | 10% | 9% |





Table 8. PCA component loadings (>0.25) of Case 10-C and Case 10-C&M.

| Variable | Case 10-C | | | | Case 10-C&M | | | | |
| --- | --- | --- | --- | --- | --- | --- | --- | --- | --- |
| | PC1 | PC2 | PC3 | PC4 | PC1 | PC2 | PC3 | PC4 | PC5 |
| GEM | | | 0.79 | | | | 0.87 | | |
| GOM | | | 0.71 | 0.33 | | | 0.51 | -0.51 | 0.38 |
| PBM | | | 0.48 | | | | 0.29 | -0.62 | |
| $O_3$ | | | 0.91 | | | | 0.87 | | |
| $SO_2$ | | | | 0.89 | | | | | 0.84 |
| $HNO_3$ | 0.34 | | | 0.83 | 0.33 | | | | 0.82 |
| $Ca^{2+}$ | 0.89 | | | | 0.89 | | | | |
| $K^+$ | 0.77 | | | | 0.77 | | | | |
| $Na^+$ | | 0.99 | | | | 0.99 | | | |
| $Mg^{2+}$ | 0.34 | 0.93 | | | 0.34 | 0.92 | | | |
| $Cl^-$ | | 0.98 | | | | 0.97 | | | |
| $NO_3^-$ | 0.79 | | | | 0.80 | | | | |
| $NH_4^+$ | 0.94 | | | | 0.94 | | | | |
| $SO_4^{2-}$ | 0.90 | | | 0.26 | 0.89 | | | | 0.26 |
| Temperature | - | - | - | - | 0.27 | | -0.52 | | 0.27 |
| Relative humidity | - | - | - | - | | | | 0.74 | -0.33 |
| Wind speed | - | - | - | - | | 0.26 | 0.52 | 0.57 | |
| Precipitation | - | - | - | - | | | | 0.76 | |
| Component | Combustion emission | Sea salt | Photochemical production of GOM | Industrial source | Combustion emission | Sea salt | Photochemical production of GOM | Hg wet deposition | Industrial source |
| Variance explained | 28% | 21% | 16% | 13% | 22% | 17% | 14% | 12% | 10% |



Table 9. General statistics of speciated Hg with different data treatment options.
a) 2009

| Hg form | Case | Percent of missing values | MDL | Percent of values <MDL | Geometric Mean | Median | Mean | Standard deviation |
|---|---|---|---|---|---|---|---|---|
| GEM (ng/m³) | 09 | 31% | | 0% | 1.37 | 1.41 | 1.39 | 0.28 |
| | 09+mean | 0% | 0.1 | 0% | 1.37 | 1.37 | 1.38 | 0.22 |
| | 09+median | 0% | | 0% | 1.38 | 1.41 | 1.39 | 0.22 |
| GOM (pg/m³) | 09 | 32% | | 73% | 0.57 | 0.42 | 1.77 | 3.98 |
| | 09+mean | 0% | | 86% | 0.57 | 0.57 | 1.39 | 3.11 |
| | 09+median | 0% | 2 | 86% | 0.51 | 0.42 | 1.34 | 3.12 |
| | 09+RM | - | | - | - | - | - | - |
| | 09Scale RM | 32% | | 16% | 3.91 | 3.35 | 5.02 | 5.49 |
| PBM (pg/m³) | 09 | 41% | 2 | 37% | 1.79 | 2.15 | 2.81 | 2.71 |
| | 09+mean | 0% | 2 | 70% | 1.79 | 1.79 | 2.39 | 2.14 |
| | 09+median | 0% | 2 | 28% | 1.93 | 2.15 | 2.53 | 2.11 |
| | 09+RM | 42% | 4 (RM) | 52% | 2.73 | 3.02 | 4.69 | 5.56 |
| | 09Scale RM | 41% | 2 | 4% | 5.52 | 6.05 | 6.19 | 3.15 |

b) 2010, MDL same as in a)

| Hg form | Case | Percent of missing values | Percent of values <MDL | Geometric Mean | Median | Mean | Standard deviation |
|---|---|---|---|---|---|---|---|
| GEM (ng/m³) | 10 | 4% | 0% | 1.33 | 1.37 | 1.34 | 0.17 |
| | 10+mean | 0% | 0% | 1.34 | 1.37 | 1.35 | 0.16 |
| | 10+median | 0% | 0% | 1.34 | 1.38 | 1.35 | 0.17 |
| | 10+RM | 4% | 0% | 1.33 | 1.37 | 1.34 | 0.17 |
| | 10ScaleRM | 4% | 0% | 1.33 | 1.38 | 1.34 | 0.17 |
| GOM (pg/m³) | 10 | 4% | 96% | 0.29 | 0.26 | 0.49 | 0.69 |
| | 10+mean | 0% | 96% | 0.27 | 0.24 | 0.43 | 0.63 |
| | 10+median | 0% | 96% | 0.27 | 0.21 | 0.43 | 0.63 |
| | 10+RM | - | - | - | - | - | - |
| | 10ScaleRM | 4% | 67% | 1.15 | 1.12 | 1.40 | 0.86 |
| PBM (pg/m³) | 10 | 4% | 51% | 1.79 | 1.92 | 2.59 | 2.67 |
| | 10+mean | 0% | 44% | 2.08 | 2.12 | 3.35 | 4.04 |
| | 10+median | 0% | 44% | 2.08 | 2.20 | 3.35 | 4.04 |
| | 10+RM | 4% | 75% | 2.16 | 2.31 | 3.08 | 2.95 |
| | 10ScaleRM | 4% | 1% | 6.15 | 6.38 | 6.75 | 3.01 |



Table 10. Impact of combining or excluding GOM and PBM on PMF factor
contributions (>15%) to Hg compounds.

| Case | Combustion emission | Industrial sulfur | Photochemistry and re-emission | Sea salt |
|---|---|---|---|---|
| Case 2009 | GOM | PBM | GEM, GOM, and PBM | |
| Case 09+RM | | RM | GEM and RM | |
| Case 09-RM | | | GEM | |
| Case 2010 | | GOM | GEM, GOM, and PBM | |
| Case 10+RM | | | GEM and RM | |
| Case 10-RM | | | GEM | |





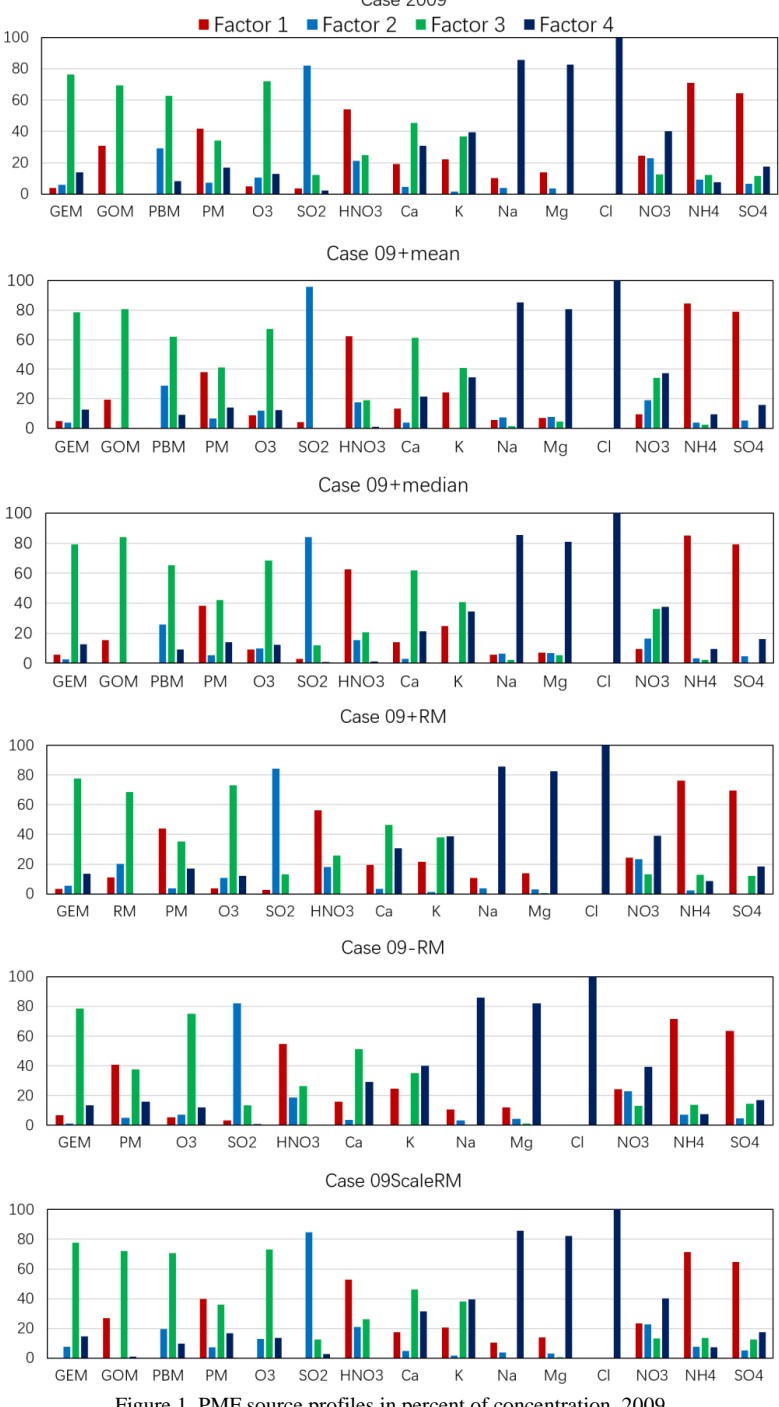

Figure 1. PMF source profiles in percent of concentration, 2009.





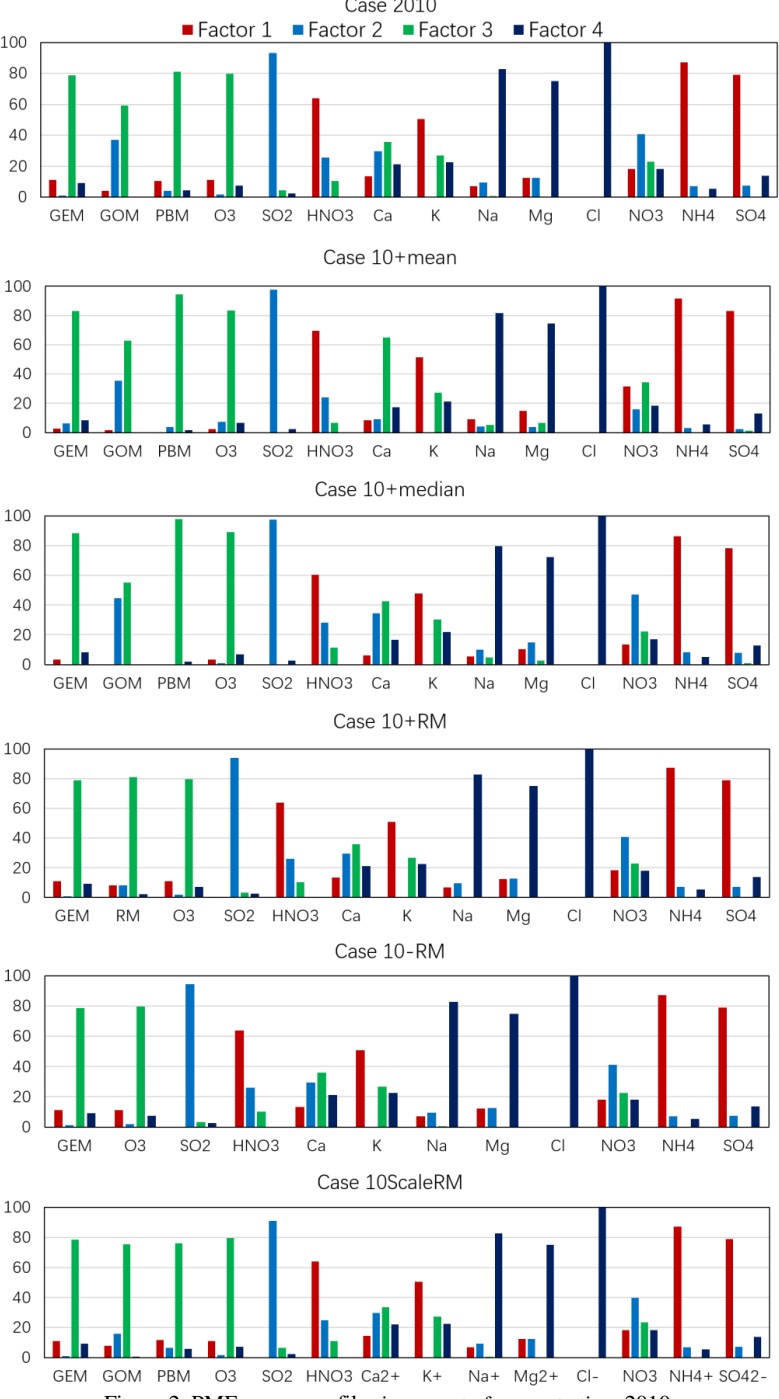

Figure 2. PMF source profiles in percent of concentration, 2010.





Figure 3. Box plot of predicted to observed concentration ratios (upper whisker- upper 25% of the distribution excluding outliers; interquartile range box - middle 50% of the data; horizontal line in the box - median; lower whisker- lower 25% of the distribution excluding outliers; ⊕ - mean).