# Peer review of "Potential sources and processes affecting speciated atmospheric mercury at Kejimkujik National Park, Canada: comparison of receptor models and data treatment methods"

_Atmospheric Chemistry and Physics, 2016_

## Referee Comment (RC1) · Anonymous Referee #1 · 1 Aug 2016

In this work, Authors aimed to identify the factors affecting ambient Hg concentrations (in the form of gaseous-GEM, oxidised-GOM and particulate-bonded-PBM) at a receptor site using PMF and PCA approaches. They also tried to summarize the similarity and differences in PMF factors and PCA components, to evaluate the PMF model performances for Hg forms, investigate the impact of meteorological parameters on PCA results, assess the sensitivity of PMF results and performance to different treatment of missing data and low concentration values of speciated Hg.

Anyway, despite the attention paid in missing/<MDL data treatments, lacks are present in describing how the PMF model was performed. In this case, a rigorous categorization of the variables is important along with the definition of the Total Variable (which

allows estimating the contribution of a chemical species to a reference variable, PM for example). In addition, a preliminary analysis of some parameters like Q, IM and IS versus the number of factors, allows the user to obtain useful information on the correct solution for PMF. All these information are not reported in the work. Finally, the authors performed an analysis of the performance of the PMF model, considering the observed/predicted forms of Hg. In this case, PMF outputs clearly shown that the model was not able to well reconstruct the variables GEM, GOM and PBM. Then in my opinion, further elaborations need in order to improve the solution and the relative stability. In conclusion, I believe that the manuscript could not be considered matured for publication in its present form considering the reasons indicated above. It could be improved with further PMF analysis. In addition, even if in this work a different data treatments have been done, the datasets analysed have already been published and this limits the novelty of the results (Chen et al., 2013). I also give some specific comments that could be useful for eventual re-writing of the paper.

Specific comments Page 2, line 64: Author could cite some example regarding works that assessed model performances of RM, such as Cesari et al., 2016 Environ Sci Pollut Res 23:15133–15148 and Belis et al. 2015 Atm Environ 123:240-250.

Page 3, line 88. A map could be useful in order to understand the sampling site position together with the sources, listed in the paper, affecting that area.

Page 4, line 111. Authors should indicate the amount of aerosol mass characterized.

Page 6, line 130. Authors should better indicate in the text the dimensions of the datasets analysed and if these dimensions respect the conditions requested in order to obtain statistically stable SA analysis. From literature, we have that these conditions are: the minimum required number N of samples N>30+0.5*(V+3) where V is the number of species considered (Henry et al., 1984 Atmos Environ18:1507-1515), and the more restrictive condition N>50+V (Thurston and Spengler, 1985 Atmos Environ19:9-25.).

Page 6, lines 176-180. Authors wrote that one of the objectives of this work is to identify the factors affecting ambient Hg concentrations using PMF model. In this sense, authors should explicit what chemical specie they used as Total Variable in PMF analysis: PM or Hg (GEM or GOM or PBM)?

Page 6, line 185: Please, give some examples of stability indexes for model runs.

Page 7, line 194. Please, indicate the dimensions of the analysed dataset and if these datasets are the same considered for PMF analysis.

Page 7, line 209. Authors should explicate why the 4-factors solution is the best solution. I am wondering if they have analysed the trend of some parameters (such as dQ, IM and IS, see Lee et al, 1999 Atmos Environ 33: 3201-3212; Viana et al., 2008 Atmos Environ 42:3820-3832; Brown et al., 2015, Sci Total Environ 518-519: 626-635) with the number of factors, from 4 to 8 for example, in order to obtain some "objective" information about the best solution. If not, please consider to perform this analysis in order to justify the choice of 4-factors solution.

Page 9, line 267. Again, what species, PM or Hg-form, has been considered as Total Variable? If the T.V. is the PM, how is possible to obtain a factor contribution for 2010?

Page 10, lines 302-307. The coefficients of determination together with Figures S1-S2 show that the model in this case is not able to reconstruct the Hg – concentrations. The reason could be different, depending on, for example, the reduced number of samples (a solution could be to merge the two datasets), or the high percentage of missing values/data lower than MDL. In my opinion, Authors should check the categorization of the variables, performed considering both the S/N value and the percentage of missing values or lower than MDL: for example, in my opinion GOM could be considered as a BAD variable. Again, the choice of a different number of factors could help in obtaining a better reconstruction. Authors should perform other runs with the aim to improve the output of the model. The same observations are for the dataset Case 2010.

[Figure]

Page 14, line 440. Referring to PC3 in table 8, how can you explain the opposite load values of GOM and Temperature? Photochemical production of GOM happens with high temperature and solar radiation, so I would imagine this variable having the same sign.
* * *

---

## Author Comment (AC1) · 13 Oct 2016

**Response to Reviewer #1 comments**

We appreciate the reviewer's constructive comments which helped us to improve the manuscript. Our point-by-point responses are provided below (in blue).

1. In this work, Authors aimed to identify the factors affecting ambient Hg concentrations (in the form of gaseous-GEM, oxidised-GOM and particulate-bonded-PBM) at a receptor site using PMF and PCA approaches. They also tried to summarize the similarity and differences in PMF factors and PCA components, to evaluate the PMF model performances for Hg forms, investigate the impact of meteorological parameters on PCA results, assess the sensitivity of PMF results and performance to different treatment of missing data and low concentration values of speciated Hg.

Anyway, despite the attention paid in missing/<MDL data treatments, lacks are present in describing how the PMF model was performed. In this case, a rigorous categorization of the variables is important along with the definition of the Total Variable (which allows estimating the contribution of a chemical species to a reference variable, PM for example). In addition, a preliminary analysis of some parameters like Q, IM and IS versus the number of factors, allows the user to obtain useful information on the correct solution for PMF. All these information are not reported in the work. Finally, the authors performed an analysis of the performance of the PMF model, considering the observed/predicted forms of Hg. In this case, PMF outputs clearly shown that the model was not able to well reconstruct the variables GEM, GOM and PBM. Then in my opinion, further elaborations need in order to improve the solution and the relative stability. In conclusion, I believe that the manuscript could not be considered matured for publication in its present form considering the reasons indicated above. It could be improved with further PMF analysis. In addition, even if in this work a different data treatments have been done, the datasets analysed have already been published and this limits the novelty of the results (Cheng et al., 2013). I also give some specific comments that could be useful for eventual re-writing of the paper.

Response: We agree with the reviewer that more analysis of the PMF model outputs would be useful. In fact, the analyses of Q, IM and IS versus the number of factors were conducted. Similarly, the percent concentrations reconstructed by all factors were monitored for each of the three Hg forms. However, these analyses were not included in the submitted manuscript. A brief description of how the optimal number of factors was determined is now included in the Methods section. Detailed analysis is presented as Supplemental Information (SI), which support the stability of PMF runs, and justify the final solution and the number of factors chosen. Several alternative PMF settings have been attempted to improve the model performance on reproducing observed GOM and PBM, including changing the category of GOM and PBM and retaining different number of factors. However, little improvement was observed in the model performance on reproducing GOM and PBM. In the revised manuscript (Results and Discussion), we have added PMF uncertainties for modeling pollutants that undergo various transformation processes, unlike the modeling of only aerosols. PMF does not account for chemical reactions that may occur as it travels from source to receptor.

In the SI, we have added the following: "The number of PMF factors needs to be chosen according to the understanding of the sources impacting the samples utilized. When the background information is not enough to determine the number of factors, several methods could be used to determine the range of the number of the factors. The maximum individual column mean (IM) and the maximum individual column standard deviation (IS) of the scaled residual matrix can be used to identify the range of the number of factors. IM and IS will show a drastic drop when the number of factors increase up to a critical value (Lee et al., 1999). The optimal number of factors should be no less than the critical value. The trend of dQ also provides useful information on deciding the number of factors. When dQ becomes small as the number of factor increases, there might be too many factors (Hopke, 2000; Brown et al., 2015). Runs with different numbers of factors in the range determined by IM, IS, and dQ should be conducted. The model performance and the interpretability of factors in each run should be evaluated. The optimal PMF solution should be a compromise of those indexes and the interpretability of the factor profiles and their comparability with those from the literature (Belis et al, 2015 b; Cesari et al., 2016).

The IM, IS, and Q values are provided in the SI of the revised manuscript as follows. "The IM and IS were calculated to determine the number of factors. The IM and IS dropped dramatically in 2009 when the number of factor increased to 3 (Figure S1). In the line plot of Q(Robust) and Q(true) vs number of factors (Figure S2), no significant decreases were found when the number of factors is larger than 5 in 2009. Therefore, the PMF was run using the number of factors from 3 to 5 in 2009. In 2010, the decrease of IS value was gradual while the IM value experienced a drastic drop when the number of factors increased to 3 (Figure S3). The trend of the Q(Robust) and Q(True) in 2010 is similar to 2009 (Figure S4). Therefore, the PMF runs with the number of factors from 3 to 5 were also conducted in 2010. The number of the factors selected (4) is a compromise of the trends of these indexes and the physical meanings of the factors obtained following Cesari et al. (2016). A detailed comparison of the physical meanings of different solutions can be found in Liao (2016)."

[Figure]

Figure S1: IM and IS vs number of PMF factors in 2009.

[Figure]

Figure S2: Q(Robust) and Q(true) vs number of PMF factors in 2009.

[Figure]

Figure S3: IM and IS vs number of PMF factors in 2010.

[Figure]

Figure S4: Q(Robust) and Q(true) vs number of PMF factors in 2010.

The same dataset was used in a PCA study (Cheng et al., 2013), while this paper focuses on PMF and comparison between PMF and PCA results. Furthermore, the variables used, treatment of missing data and number of component are different. In Cheng et al. (2013)'s study, pairwise exclusion was used to make the full use of the dataset. The marine tracing species were excluded in 2009 while $SO_2$, $HNO_3$, and all ions were excluded in 2010 because they were not related with mercury. However, listwise exclusion and all species were used in order to be compared with the PMF results in this study. The method used to retain the number of components for further analysis was different. Fixed number (4 and 3 for 2009 and 2010, respectively) of components was retained in Cheng et al. (2013)'s study but the Kaiser criterion (eigenvalue>1) was used to retain the number of components in this study. Those are presented in Table 4. All these differences could result in the differences in the PCA results. In short, the comparison of the results suggests that the PCA results are sensitive to the input parameters. In terms of the differences in the results, four components were extracted in Cheng et al. (2013)'s study in 2009. Three out of four components, including Combustion/Industrial Source, Gas-particle partitioning of Hg, and Gas-phase Oxidation of Hg, were similar as the components in Case 09-C&M. The component loadings of the components Combustion/industrial Source and Gas-phase Oxidation of Hg in 2009 were similar in this study and in Cheng et al. (2013)'s study. The component loadings of the components Condensation on Particles in Winter (Cheng et al., 2013) and Gas-particle Partitioning of Hg (this study) in 2009 were very different. Only the negative association between temperature and PBM was the same between Cheng et al. (2013)'s study and this study. Three components were extracted by Cheng et al. (2013) in 2010. However, none of the major variables of these three components is similar to the five components identified in this study in Case 10-C&M. In a PM10 source apportionment study using PMF and PCA at three European sites (Cesari et al., 2016), the authors reported that PCA results are more sensitive to the air contaminants present as input variables. In the revised manuscript, a reference has been added (Liao, 2016) for a detailed comparison of PCA results in this study and that in Cheng et al. (2013).

**Specific comments**

2. Page 2, line 64: Author could cite some example regarding works that assessed model performances of RM, such as Cesari et al., 2016 Environ Sci Pollut Res 23:15133–15148 and Belis et al. 2015 Atm Environ 123:240-250.

Response: Thank you for your suggestion, those papers are included in the revised manuscript, Method as well as Results and Discussion sections.

3. Page 3, line 88. A map could be useful in order to understand the sampling site position together with the sources, listed in the paper, affecting that area.

Response: A map (Figure 1) has been added as suggested, including the sampling site, all Hg sources and all major sources of NOx and $SO_2$ in Nova Scotia.

[Figure]

Figure 1: Map showing the locations of sampling site (▲), the top 19 $SO_2$ or $NO_x$ point sources (average of 2009 and 2010) (★), and all mercury point sources in 2009 and 2010 (⬡), in Nova Scotia.

4. Page 4, line 111. Authors should indicate the amount of aerosol mass characterized.

Response: The total aerosol mass characterized in each year has been reported in the Methods section of the revised manuscript. The mass of total ions in 2009 accounted for 80% of the PM mass. The total ions were added in Table 1 and Table 2 for 2009 and 2010, respectively.

5. Page 6, line 130. Authors should better indicate in the text the dimensions of the datasets analysed and if these dimensions respect the conditions requested in order to obtain statistically stable SA analysis. From literature, we have that these conditions are: the minimum required number N of samples N>30+0.5*(V+3) where V is the number of species considered (Henry et al., 1984 Atmos Environ18:1507-1515), and the more

restrictive condition N>50+V (Thurston and Spengler, 1985 Atmos Environ19:9-25.).

Response: Dimension of datasets has been incorporated in Tables 3 and 4 as suggested. We have also added in the main body the conditions requested in order to obtain statistically stable source apportionment results (Henry et al., 1984; Thurston and Spengler, 1985), and stated that our datasets meet the more restrictive requirement by Thurston and Spengler (1985) in both years, by a margin of 90-300 in 2009 and 216-300 in 2010.

6. Page 6, lines 176-180. Authors wrote that one of the objectives of this work is to identify the factors affecting ambient Hg concentrations using PMF model. In this sense, authors should explicit what chemical specie they used as Total Variable in PMF analysis: PM or Hg (GEM or GOM or PBM)?

Response: This is now clarified in the Methods section as suggested. We did not use total variable because this study focused on speciated mercury. Commonly used total variables include PM as pointed out by the reviewer. However, input variables in this study include both PM ions and gaseous pollutants. Furthermore, a total variable should be set to "weak" as recommended by USEPA (US EPA, 2014) thus may not have much impact on the PMF results.

7. Page 6, line 185: Please, give some examples of stability indexes for model runs.

Response: Included now in Supplemental Information. As pointed out in the PMF User's Guide (US EPA, 2014), when the Q (robust) values over several runs are highly variable, the stability of the result is poor. In this study, the differences of the Q (Robust) value between different runs were all smaller than 5 indicating that the results were quite stable in both years.

8. Page 7, line 194. Please, indicate the dimensions of the analysed dataset and if these datasets are the same considered for PMF analysis.

Response: Included in text now under the Methods section, as suggested. The dimensions of the reference cases in PMF model and PCA are the same. After including the meteorological parameters in PCA input, the dimensions of the input data are slightly smaller.

9. Page 7, line 209. Authors should explicate why the 4-factors solution is the best solution. I am wondering if they have analysed the trend of some parameters (such as dQ, IM and IS, see Lee et al, 1999 Atmos Environ 33: 3201-3212; Viana et al., 2008 Atmos Environ 42:3820-3832; Brown et al., 2015, Sci Total Environ 518-519: 626-635) with the number of factors, from 4 to 8 for example, in order to obtain some "objective" information about the best solution. If not, please consider to perform this analysis in order to justify the choice of 4-factors solution.

Response: Included now as Supplemental Information, please see response to comment #1.

10. Page 9, line 267. Again, what species, PM or Hg-form, has been considered as Total

Variable? If the T.V. is the PM, how is possible to obtain a factor contribution for 2010?

Response: This is now clarified as suggested; please see response to comment #6.

11. Page 10, lines 302-307. The coefficients of determination together with Figures S1-S2 show that the model in this case is not able to reconstruct the Hg – concentrations. The reason could be different, depending on, for example, the reduced number of samples (a solution could be to merge the two datasets), or the high percentage of missing values/data lower than MDL. In my opinion, Authors should check the categorization of the variables, performed considering both the S/N value and the percentage of missing values or lower than MDL: for example, in my opinion GOM could be considered as a BAD variable. Again, the choice of a different number of factors could help in obtaining a better reconstruction. Authors should perform other runs with the aim to improve the output of the model. The same observations are for the dataset Case 2010.

Response: Included now as supplemental Information. The S/N ratio was not used in this study because the uncertainties of all variables were set to a fixed fraction of concentrations as suggested in the PMF user's guide. Because this study focuses on speciated mercury, all three mercury forms should be included in the input. Also, categorizing GOM and PBM to "weak" would have a similar reproduction of GOM and PBM concentrations compared to the cases categorizing them as "strong". Therefore, they are categorized as strong in this study. Different numbers of factors were also analyzed and the 4-factor result had the best interpretability. Therefore, 4 was used as the number of factors.

12. Page 14, line 440. Referring to PC3 in table 8, how can you explain the opposite load values of GOM and Temperature? Photochemical production of GOM happens with high temperature and solar radiation, so I would imagine this variable having the same sign.

Response: This is now clarified as suggested in the Results (section 3.2). "The additional negative loading of temperature (-0.52, Table 8) and positive loading of wind speed (0.52, Table 8) in major variables may indicate colder air flows from the north containing more $O_3$ and GOM (Cheng et al., 2013). This is reasonable because Hg sources in Nova Scotia were mainly located north of the sampling site (Figure 1)."

References

Belis, C. A., Pernigotti, D., Karagulian, F., Pirovano, G., Larsen, B. R., Gerboles, M., and Hopke, P. K.: A New Methodology to Assess the Performance and Uncertainty of Source Apportionment Models in Intercomparison Exercises, Atmos. Environ., 119, 35-44, 10.1016/j.atmosenv.2015.08.002, 2015a.

Belis, C. A., Karagulian, F., Amato, F., Almeida, M., Artaxo, P., Beddows, D. C. S., Bernardoni, V., Bove, M. C., Carbone, S., Cesari, D., Contini, D., Cuccia, E., Diapouli, E., Eleftheriadis, K., Favez, O., El-Haddad, I., Harrison, R. M., Hellebust, S., Hovorka, J., Jang, E., Jorquera, H., Kammermeier, T., Karl, M., Lucarelli, F., Mooibroek, D., Nava, S., Nøjgaard, J. K., Paatero, P., Pandolfi, M., Perrone, M. G., Petit, J. E., Pietrodangelo, A., Pokorná, P., Prati, P., Prevot, A. S. H., Quass, U., Querol, X., Saraga, D., Sciare, J., Sfetsos,

A., Valli, G., Vecchi, R., Vestenius, M., Yubero, E., and Hopke, P. K.: A New Methodology to Assess the Performance and Uncertainty of Source Apportionment Models II: The Results of Two European Intercomparison Exercises, Atmos. Environ., 123, 240-250, 10.1016/j.atmosenv.2015.10.068, 2015b.

Brown, S. G., Eberly, S., Paatero, P., and Norris, G. A.: Methods for Estimating Uncertainty in PMF Solutions: Examples with Ambient Air and Water Quality Data and Guidance on Reporting PMF Results, Sci. Total Environ., 518-519, 626-635, dx.doi.org/10.1016/j.scitotenv.2015.01.022, 2015.

Cesari, D., Amato, F., Pandolfi, M., Alastuey, A., Querol, X., and Contini, D.: An Inter-comparison of PM10 Source Apportionment Using PCA and PMF Receptor Models in Three European Sites, Environment Science and Pollution Research, 23, 15133-15148, 10.1007/s11356-016-6599-z, 2016.

Cheng, I., Zhang, L., Blanchard, P., Dalziel, J., Tordon, R., Huang, J., and Holsen, T. M.: Comparisons of Mercury Sources and Atmospheric Mercury Processes between a Coastal and Inland Site, J, Geophys. Res-Atmos., 118, 2434-2443, 10.1002/jgrd.50169, 2013.

Henry, R. C., Lewis, C. W., Hopke, P. K., and Williamson, H. J.: Review of Receptor Model Fundamentals, Atmos. Environ., 18, 1507-1515, 10.1016/0004-6981(84)90375-5 1984.

Hopke, P. K.: A Guide to Positive Matrix Factorization, available at: https://www3.epa.gov/ttnamti1/files/ambient/pm25/workshop/laymen.pdf (last access: May 30,2016), 2000.

Lee, E., Chan, C. K., and Paatero, P.: Application of Positive Matrix Factorization in Source Apportionment of Particulate Pollutants in Hong Kong, Atmos. Environ., 33, 3201-3212, 10.1016/S1352-2310(99)00113-2, 1999.

Liao, Y.: Analysis of Potential Sources and Processes Affecting Ambient Speciated Mercury Concentrations at Kejimkujik National Park, Nova Scotia, Master's thesis, Department of Civil and Environmental Engineering, University of Windsor, Windsor, Ontario, Canada. 2016.

Thurston, G. D., and Spengler, J. D.: A Quantitative Assessment of Source Contributions to Inhalable Particulate Matter Pollution in Metropolitan Boston, Atmos. Environ., 19, 9-25, doi:10.1016/0004-6981(85)90132-5, 1985.

US Environmental Protection Agency (US EPA): EPA Positive Matrix Factorization (PMF) 5.0 Fundamentals and User Guide, available at: https://www.epa.gov/sites/production/files/2015-02/documents/pmf_5.0_user_guide.pdf (last access: May 30, 2016), 2014.

---

## Referee Comment (RC2) · Anonymous Referee #2 · 8 Nov 2016

This study used two-year Hg measurements (Tekran) with other air pollutants at Kejimkujik national park in Canada, and applied factor analysis (PMF) and principle component analysis to understand Hg sources and its related atmospheric processes. Overall, this is a well written article and easy to be followed paper. A very similar paper was published couple years ago; however, I understand the authors applied PMF as an additional analysis, and investigated how model setting impacts receptor modeling. There are couple things I would like suggest to the authors to look into detail: 1) Wang et al., 2013 Chemosphere and Huang et al., 2010 ES&T have compared results from PCA and PMF using Hg related concentrations at Rochester, NY using similar data set. PCA and PMF comparisons using aerosol data have been discussed in detail in

previous studies (Paatero and Tapper, 1994; Environmentrics, 1994). 2) This study and Cheng et al., 2013 are using similar data set with similar results. What is new that we can learn from this study? After reading the abstract, I think the one new thing to the global Hg research group is the difference between 2009 and 2010. I suggest the authors should focus on these important things instead of repeating what we already knew or has been published on journals. I suggest a minor revision before ACP can accept this article. The specific comments are listed below:

In abstract, the authors focus on comparison of result from different models; however, the title looks more like a straight source paper, suggest to modify either the title or abstract. After reading this paper, one selling point is both model can capture the significant reduction of Hg and SO2 from 2009 to 2010. However, it is not mentioned in the abstract. Line 46-64, the most important difference between PMF (as a factor analysis) and PCA is the different concepts of these two receptor models, PMF constrains factor loadings and factor scores to nonnegative values and thereby minimizes the ambiguity caused by rotating factors. I suggest the authors dig this into detail and include the information there. Line 60-62, Wang et al., 2013 Chemosphere and Huang et al., 2010 ES&T have done the comparison between PMF and PCA using Hg data. Line 62-64, PMF has been applied to aerosol and evaluated in plenty previous studies, Belis et al., 2013 is a good article to start. Line 120-123, many people using GOM and PBM to do advanced statistical analysis, the biggest problem is how to handle missing and BMDL data. I look into table 1, a large portion of GOM/PBM is missing or BMDL. I understand that is the limitation of using statistical modeling on Hg data, but it will skew data distribution significantly. Line 142-143, after reading the entire paper, I still don't fully understand these cases. Line 165, the authors used manufacture method detection limit. However, this can vary with locations and time, can the authors also talk about real MDL for Tekran system at this site? Line 166, why MDL for RM is 4 ng m-3? MDL is defined as 3 standard deviation of blanks, that could be the upper bound of MDL for RM, but if you look into distribution sum, that might be lower. Line 181, how did the authors select number of factor in PMF? In general, we look into Q

and the variation of Q and number of factor. Line 238, Is this possible only due to biomass burning + soil emissions? We will see high ozone with biomass combustion, and it matches to all these increase for GEM, GOM, PBM, ozone. Does this happen in summer or winter, if you look into detail time series factor profiles, the authors should be able to figure this out. Line 302, I don't suggest using these analyses to predict GOM and PBM concentrations, as discussed above, a large portion of GOM and PBM is missing and BMDL.

Table 7, in the column title they are Case 10, but I think they should be 9.

[Figure]

---

## Referee Comment (RC3) · Anonymous Referee #3 · 10 Nov 2016

The receptor models for source apportionment of atmospheric mercury are of great importance. This study applied PMF and PCA on the data of speciated mercury and other tracers from a coastal observation site. Different methods of data processing were conducted for comparison. The comparison between PMF and PCA as well as between the two monitoring years was also performed. Advantages and disadvantages of the two receptor models were discussed. Overall, it is an important exploration of receptor models applying to atmospheric mercury studies. Elaborations on some key points are still needed. Therefore, I suggest the manuscript be accepted for the publication on Atmospheric Chemistry and Physics after major revision. Here are some specific comments: 1. Lines 46–64: This paragraph could use more literatures. Although the authors have reviewed the receptor model studies on atmospheric mercury in their previous paper (Cheng et al., 2015), examples on the applications of PMF and PCA are still needed in the introduction of this paper, not limited to atmospheric mercury. For example, Gibson et al. (2015) compared the four receptor models for PM2.5 source apportionment in Halifax. Some models could be more suitable for PM2.5 than for mercury. The authors could provide more proof on the merits and drawbacks of PMF and PCA when applied to atmospheric mercury. 2. Line 56: How do the authors define "qualitative" here? Aren't the loadings of the PCA method quantitative? To my understanding, PMF describes the contributions of one parameter in different factors, while PCA describes the contributions of different parameters in one PC. The quantitative contribution of each PC to the receptor can be reflected by the "variance explained" (in Table 7 and 8). 3. Section 2.1: A map of the observation site with locations of the emission sources listed in Table S1 and a brief description of the meteorological conditions would be useful. This information could be referred to in the discussion part to verify the results from the receptor models. 4. Lines 113–114: Is there any specific reason why the authors averaged the original data to daily values? If the original data is hourly or 3-hr, it should be possible to obtain 3-hr, 6-hr or 12-hr averages, which could result in a larger database for PMF and PCA. Isn't it better? 5. Line 145: The expression "resultant PMF result" seems repetitive. 6. Lines 192–193: Since the PCA analysis has already been conducted in Cheng et al. (2013), I think the current title of the manuscript is inappropriate. It could give the readers misimpression that this is partially repeated from the previous study. To my understanding, the methodology of this study is the novelty of this paper. Therefore, it is better to embody the methodology in the title. 7. Line 203: Have the authors checked the inter-correlations between any two of the major PCs? Varimax is an orthogonal rotation method, which requires the PCs to be independent on each other. This validation process for the applicability of the Varimax rotation could be mentioned here. 8. Table 5 and Figure 1-2: NO3 in Table 5 should be NO3-. All the "+" and "-" signs cannot be omitted in Figure 1-2. NO3 and NO3- stand for different compounds. 9. Line 212: From the context (Lines

267–268), Combustion Emission include both coal combustion and biomass burning? It is better to mention it here. Does open biomass burning or wildfires included in F1? 10. Line 232: Can the authors specify what types of sources could be Industrial Sulfur? Non-ferrous metal melting? What could be the possible Industrial Sulfur sources in this region? 11. Line 238: The authors mentioned biomass combustion in this part while the name of Factor 3 is Photochemical Process and Re-emission of Hg. Why is it necessary for the biomass combustion to be related to Re-emission? Is it possible that F1 is composed of coal combustion and controlled biomass combustion which are usually mixed from regional sources while F3 is composed of mineral dust and open biomass burning/wildfires which are usually mixed in long-range transport? 12. Table 6: The performance of 2009 GOM and 2010 PBM is poor to me. I don't think the previous discussion linked to these two parts can be validated. Is it possible to improve the model performance by using the 3-hr or 6-hr averages instead of daily averages to increase the size of the database?

Reference: Gibson, M. D., Haelssig, J., Pierce, J. R., Parrington, M., Franklin, J. E., Hopper, J. T., Li, Z., and Ward, T. J.: A comparison of four receptor models used to quantify the boreal wildfire smoke contribution to surface PM2.5 in Halifax, Nova Scotia during the BORTAS-B experiment, Atmos. Chem. Phys., 15(2), 815–827, 2015.

---

## Author Response (AR1)

**Response to Reviewers #2 and #3 comments**

We appreciate both reviewers' constructive comments which helped us to improve the manuscript. Our point-by-point responses are provided below (in blue). The manuscript has been revised to reflect comments and suggestions by all three reviewers. Track change, yellow highlight (changes in tables and figures), or blue (added Figure 1 & SI sections) were used in the marked-up manuscript uploaded online.

**Anonymous Referee #2**

This study used two-year Hg measurements (Tekran) with other air pollutants at Kejimkujik national park in Canada, and applied factor analysis (PMF) and principle component analysis to understand Hg sources and its related atmospheric processes. Overall, this is a well written article and easy to be followed paper. A very similar paper was published couple years ago; however, I understand the authors applied PMF as an additional analysis, and investigated how model setting impacts receptor modeling.

There are couple things I would like suggest to the authors to look into detail:

1) Wang et al., 2013 Chemosphere and Huang et al., 2010 ES&T have compared results from PCA and PMF using Hg related concentrations at Rochester, NY using similar data set. PCA and PMF comparisons using aerosol data have been discussed in detail in previous studies (Paatero and Tapper, 1994; Environmentrics, 1994).

These three papers have been included in the revised manuscript, "Comparisons of results of receptor models for PM source apportionment have been reported, e.g. Paatero and Tapper (1994), Viana et al. (2008), Belis et al. (2013), and Gibson et al. (2015). To date, PCA and PMF have been applied to atmospheric Hg and other air pollutants in Toronto (Canada) (Cheng et al., 2009) and in Rochester, New York (USA) (Huang et al., 2010; Wang et al., 2013). However, both the Toronto and Rochester studies lacked a thorough comparison of the PMF and PCA results."

2) This study and Cheng et al., 2013 are using similar data set with similar results. What is new that we can learn from this study? After reading the abstract, I think the one new thing to the global Hg research group is the difference between 2009 and 2010. I suggest the authors should focus on these important things instead of repeating what we already knew or has been published on journals. I suggest a minor revision before ACP can accept this article. The specific comments are listed below: In abstract, the authors focus on comparison of result from different models; however, the title looks more like a straight source paper, suggest to modify either the title or abstract.

Good suggestion about the tile, it has been revised as "Potential sources and processes affecting speciated atmospheric mercury at Kejimkujik National Park, Canada: comparison of receptor models and data treatment methods."

The same dataset was used in a PCA study (Cheng et al., 2013), while this paper focuses on PMF and comparison between PMF and PCA results. Furthermore, the variables used, treatment of missing data and number of component are different. In Cheng et al. (2013)'s study, pairwise exclusion was used to make the full use of the dataset. The marine tracing species were excluded in 2009 while SO2, HNO3, and all ions were excluded in 2010 because they were not related with mercury. However, listwise exclusion and all species were used in order to be compared with the PMF results in this study. The method used to retain the number of components for further analysis was different. Fixed number (4 and 3 for 2009 and 2010, respectively) of components was retained in Cheng et al. (2013)'s study but the Kaiser criterion (eigenvalue>1) was used to retain the number of components in this study. Those are presented in Table 4. All these differences could result in the differences in the PCA results. In short, the comparison of the results suggests that the PCA results are sensitive to the input parameters. In terms of the differences in the results, four components were extracted in Cheng et al. (2013)'s study in 2009. Three out of four components, including Combustion/Industrial Source, Gas-particle partitioning of Hg, and Gas-phase Oxidation of Hg, were similar as the components in Case 09-C&M. The component loadings of the components Combustion/industrial Source and Gas-phase Oxidation of Hg in 2009 were similar in this study and in Cheng et al. (2013)'s study. The component loadings of the components Condensation on Particles in Winter (Cheng et al., 2013) and Gas-particle Partitioning of Hg (this study) in 2009 were very different. Only the negative association between temperature and PBM was the same between Cheng et al. (2013)'s study and this study. Three components were extracted by Cheng et al. (2013) in 2010. However, none of the major variables of these three components is similar to the five components identified in this study in Case 10-C&M. In a PM10 source apportionment study using PMF and PCA at three European sites (Cesari et al., 2016), the authors reported that PCA results are more sensitive to the air contaminants present as input variables. In the revised manuscript, a reference has been added (Liao, 2016) for a detailed comparison of PCA results in this study and that in Cheng et al. (2013).

After reading this paper, one selling point is both model can capture the significant reduction of Hg and SO2 from 2009 to 2010. However, it is not mentioned in the abstract.

Agree, the following sentence was added in section 3.1, "Moreover, the long term effects of emission reductions on Hg concentrations and source contributions should be investigated."

Line 46-64, the most important difference between PMF (as a factor analysis) and PCA is the different concepts of these two receptor models, PMF constrains factor loadings and factor scores to nonnegative values and thereby minimizes the ambiguity caused by rotating factors. I suggest the authors dig this into detail and include the information there.

Agree, in the revised manuscript, we have referenced more papers for detailed comparison of the PMF and PCA approaches in the Introduction section. "Various receptor models have been used

to identify the sources and processes affecting air pollutant levels. Strengths and weaknesses of some receptor models have been reported previously (e.g. Watson et al., 2008, Viana et al., 2008; Belis et al., 2013)."

Line 60-62, Wang et al., 2013 Chemosphere and Huang et al., 2010 ES&T have done the comparison between PMF and PCA using Hg data.

In the revised manuscript, we have included both studies "Comparisons of results of receptor models for PM source apportionment have been reported, e.g. Paatero and Tapper (1994), Viana et al. (2008), Belis et al. (2013), and Gibson et al. (2015). To date, PCA and PMF have been applied to atmospheric Hg and other air pollutants in Toronto (Canada) (Cheng et al., 2009) and in Rochester, New York (USA) (Huang et al., 2010; Wang et al., 2013). However, both the Toronto and Rochester studies lacked a thorough comparison of the PMF and PCA results."

Line 62-64, PMF has been applied to aerosol and evaluated in plenty previous studies, Belis et al., 2013 is a good article to start.

Agree, the article of Belis et al. (2013) has been included in the revised manuscript, see reply to comment 1).

Line 120-123, many people using GOM and PBM to do advanced statistical analysis, the biggest problem is how to handle missing and BMDL data. I look into table 1, a large portion of GOM/PBM is missing or BMDL. I understand that is the limitation of using statistical modeling on Hg data, but it will skew data distribution significantly.

We agree with the reviewer that when a large portion of GOM/PBM is missing or BMDL, it will skew the data distribution significantly. Data treatment techniques such as those used in this study may not improve the data distribution but may lead to improved model performance.

Line 142-143, after reading the entire paper, I still don't fully understand these cases.

Please see Tables 3 and 4 for a summary of the various cases which describes the different input variables and treatment of missing data.

Line 165, the authors used manufacture method detection limit. However, this can vary with locations and time, can the authors also talk about real MDL for Tekran system at this site?

We agree with the reviewer that study period and location specific MDLs would be useful. However, such MDLs for GOM and PBM cannot be accurately determined at this site and the majority of other monitoring sites using the Tekran system. This is primarily due to the lack of calibration standards for GOM and PBM as pointed out by Jaffe et al. (2014) and Gustin et al. (2015) and a lack of technologies to determine individual species of GOM (Jones et al., 2016). Therefore we used manufacture method detection limits for GOM and PBM for the purpose of identifying concentrations values with large uncertainties.

Line 166, why MDL for RM is 4 ng m-3? MDL is defined as 3 standard deviation of blanks, that could be the upper bound of MDL for RM, but if you look into distribution sum, that might be lower.

Added in the revised manuscript, "For RM, the MDL was assumed to be 4 $pg/m^3$, which is a summation of MDLs of GOM and PBM (2 $pg/m^3$ each)."

Line 181, how did the authors select number of factor in PMF? In general, we look into Q and the variation of Q and number of factor.

We agree with the reviewer that more analysis of the PMF model outputs would be useful. In fact, the analyses of Q, IM and IS versus the number of factors were conducted. Similarly, the percent concentrations reconstructed by all factors were monitored for each of the three Hg forms. However, these analyses were not included in the submitted manuscript. A brief description of how the optimal number of factors was determined is now included in the Methods section. Detailed analysis is presented as Supplemental Information (SI), which support the stability of PMF runs, and justify the final solution and the number of factors chosen.

2) Line 238, Is this possible only due to biomass burning + soil emissions? We will see high ozone with biomass combustion, and it matches to all these increase for GEM, GOM, PBM, ozone. Does this happen in summer or winter, if you look into detail time series factor profiles, the authors should be able to figure this out.

An examination of the time series factor profiles revealed that model-reproduce $K^+$, $O_3$ and GEM, GOM, PBM concentrations (in this factor) were rather smooth without any episodes of high O3, K+, and Hg forms identified. The relatively stable patterns of K+ and GEM suggest re-emission of GEM while GOM was high in spring with elevated $K^+$, $O_3$, indicating enhanced photochemistry. In the revised manuscript, we have added, "An examination of the time series factor profiles revealed that model-reproduced $K^+$, $O_3$ and GEM, GOM, PBM concentrations (in this factor) were rather smooth. The impact of biomass burning seems to be small in this factor due to a lack of high $K^+$, $O_3$, and Hg concentration periods or episodes identified. The relatively stable patterns of $K^+$ and GEM suggest re-emission of GEM while GOM was high in spring with elevated $O_3$, indicating enhanced photochemistry."

3) Line 302, I don't suggest using these analyses to predict GOM and PBM concentrations, as discussed above, a large portion of GOM and PBM is missing and BMDL.

Agree, we have replaced the word "predicted" with "model-reproduced" or "reproduced" throughout the revised manuscript.

4) Table 7, in the column title they are Case 10, but I think they should be 9.

The reviewer is correct; it should be "Case 9" in the column heading.

**Anonymous Referee #3**

The receptor models for source apportionment of atmospheric mercury are of great importance. This study applied PMF and PCA on the data of speciated mercury and other tracers from a coastal observation site. Different methods of data processing were conducted for comparison. The comparison between PMF and PCA as well as between the two monitoring years was also performed. Advantages and disadvantages of the two receptor models were discussed. Overall, it is an important exploration of receptor models applying to atmospheric mercury studies. Elaborations on some key points are still needed. Therefore, I suggest the manuscript be accepted for the publication on Atmospheric Chemistry and Physics after major revision. Here are some specific comments:

1. Lines 46–64: This paragraph could use more literatures. Al-though the authors have reviewed the receptor model studies on atmospheric mercury in their previous paper (Cheng et al., 2015), examples on the applications of PMF and PCA are still needed in the introduction of this paper, not limited to atmospheric mercury. For example, Gibson et al. (2015) compared the four receptor models for $PM_{2.5}$ source apportionment in Halifax. Some models could be more suitable for $PM_{2.5}$ than for mercury. The authors could provide more proof on the merits and drawbacks of PMF and PCA when applied to atmospheric mercury.

Agree, we have included more papers in the revised manuscript, including the following: "Various receptor-based models have been used to identify the sources and processes affecting air pollutant levels. Strengths and weaknesses of some receptor models have been reported previously (e.g. Watson et al., 2008, Viana et al., 2008; Belis et al., 2013)." "Comparisons of results of receptor models for PM source apportionment have been reported, e.g. Paatero and Tapper (1994), Viana et al. (2008), Belis et al. (2013), and Gibson et al. (2015). To date, PCA and PMF have been applied to atmospheric Hg and other air pollutants in Toronto (Canada) (Cheng et al., 2009) and in Rochester, New York (USA) (Huang et al., 2010; Wang et al., 2013). However, both the Toronto and Rochester studies lacked a thorough comparison of the PMF and PCA results."

2. Line 56: How do the authors define "qualitative" here? Aren't the loadings of the PCA method quantitative? To my understanding, PMF describes the contributions of one parameter in different factors, while PCA describes the contributions of different parameters in one PC. The quantitative contribution of each PC to the receptor can be reflected by the "variance explained" (in Table 7 and 8).

Agree, the sentence has been rephrased as "PCA can only provide qualitative assessment of sources/processes but it cannot determine the source contributions to pollutant concentrations (Hopke, 2015).". This is because the PCA loadings are not the same as the source contributions in PMF. In PCA, the loadings reflect the correlation coefficients between the variables and component, which are used to qualitatively assign the components to sources. Furthermore, the variance explained by each component in PCA is also not equivalent to source contributions to receptor measurements; it is only a measure of how well the component can explain the

variability in the dataset.  The PMF factor contributions actually quantify the contribution of each source to atmospheric pollutant concentrations.

3. Section 2.1: A map of the observation site with locations of the emission sources listed in Table S1 and a brief description of the meteorological conditions would be useful. This information could be referred to in the discussion part to verify the results from the receptor models.

A map (Figure 1) has been included, also added statistics of meteorological parameters in Tables 1-2 with the following statements, "The weather conditions were similar in the two years, with an annual mean relative humidity of 88% and 87% in 2009 and 2010 respectively, moderate wind speeds (4.7 km/h and 4.4 km/h), but a higher precipitation amount  (1597 mm/yr vs. 1480 mm/yr) and a lower temperature (6.6℃ vs. 8.1℃) in 2009 than 2010."

Table 1. General statistics of daily air pollutant concentrations (in µg/m$^3$ unless otherwise noted) and meteorological parameters in 2009.

| Compound | Percent of missing values | Method detection limit (MDL) | Percent of values <MDL | Geometric Mean | Median | Mean | Standard deviation | Coefficient of variability (%) |
|---|---|---|---|---|---|---|---|---|
| GEM (ng/m$^3$) | 31% | 0.1 | 0% | 1.37 | 1.41 | 1.39 | 0.26 | 18.7 |
| GOM (pg/m$^3$) | 32% | 2 | 78% | 0.57 | 0.42 | 1.77 | 3.70 | 209 |
| PBM (pg/m$^3$) | 41% | 2 | 48% | 1.78 | 2.15 | 2.81 | 2.72 | 96.8 |
| PM | 20% | 1 | 9% | 2.71 | 2.91 | 3.44 | 2.49 | 72.4 |
| O$_3$ | 0% | 4.3 | 0% | 59.4 | 62.1 | 62.4 | 19.1 | 30.6 |
| SO$_2$ | 3% | 0.002 | 0% | 0.20 | 0.22 | 0.40 | 0.51 | 128 |
| HNO$_3$ | 3% | 0.05 | 12% | 0.13 | 0.12 | 0.19 | 0.22 | 116 |
| Ca$^{2+}$ | 1% | 0.002 | 0% | 0.05 | 0.05 | 0.06 | 0.04 | 66.7 |
| K$^+$ | 1% | 0.029 | 17% | 0.04 | 0.03 | 0.04 | 0.03 | 75.0 |
| Na$^+$ | 1% | 0.05 | 9% | 0.25 | 0.30 | 0.43 | 0.47 | 109 |
| Mg$^{2+}$ | 1% | 0.0004 | 2% | 0.04 | 0.04 | 0.06 | 0.06 | 100 |
| Cl$^-$ | 1% | 0.046 | 23% | 0.19 | 0.23 | 0.46 | 0.64 | 139 |
| NO$_3^-$ | 1% | 0.06 | 9% | 0.18 | 0.17 | 0.28 | 0.39 | 139 |
| NH$_4^+$ | 1% | 0.001 | 0% | 0.19 | 0.17 | 0.28 | 0.32 | 114 |
| SO$_4^{2-}$ | 1% | 0.05 | 0% | 0.78 | 0.76 | 1.14 | 1.27 | 111 |
| Total ions | 1% | - | - | 2.13 | 2.05 | 2.76 | 2.23 | 81 |
| Temperature (°C) | 0% | - | - | - | 7.31 | 6.64 | 9.28 | 140 |
| Relative humidity (%) | 0% | - | - | - | 87.5 | 84.5 | 12.0 | 14 |
| Wind speed (m/s) | 0% | - | - | - | 4.33 | 4.70 | 2.39 | 51 |
| Precipitation (mm/day) | 3% | - | - | - | 0.60 | 4.50 | 10.0 | 222 |

Table 2. General statistics of daily air pollutant concentrations (in µg/m³ unless otherwise noted) and meteorological parameters in 2010, MDL same as in Table 1.

| Compound | Percent of missing values | Percent of values <MDL | Geometric Mean | Median | Mean | Standard deviation | Coefficient of variability (%) |
|---|---|---|---|---|---|---|---|
| GEM (ng/m³) | 4% | 0% | 1.34 | 1.38 | 1.35 | 0.17 | 12.6 |
| GOM (pg/m³) | 4% | 96% | 0.27 | 0.21 | 0.44 | 0.64 | 145 |
| PBM (pg/m³) | 4% | 46% | 2.08 | 2.20 | 3.40 | 4.13 | 121 |
| $O_3$ | 1% | 0% | 62.2 | 63.4 | 64.5 | 16.6 | 25.7 |
| $SO_2$ | 19% | 1% | 0.10 | 0.13 | 0.23 | 0.31 | 135 |
| $HNO_3$ | 19% | 25% | 0.10 | 0.10 | 0.18 | 0.22 | 122 |
| $Ca^{2+}$ | 19% | 0% | 0.04 | 0.04 | 0.07 | 0.13 | 186 |
| $K^+$ | 19% | 46% | 0.04 | 0.03 | 0.06 | 0.07 | 117 |
| $Na^+$ | 19% | 16% | 0.20 | 0.24 | 0.40 | 0.53 | 133 |
| $Mg^{2+}$ | 19% | 0 % | 0.03 | 0.04 | 0.05 | 0.06 | 120 |
| $Cl^-$ | 19% | 27% | 0.14 | 0.15 | 0.46 | 0.83 | 180 |
| $NO_3^-$ | 19% | 21% | 0.14 | 0.13 | 0.25 | 0.36 | 144 |
| $NH_4^+$ | 19% | 0% | 0.16 | 0.15 | 0.30 | 0.57 | 190 |
| $SO_4^{2-}$ | 19% | 0% | 0.69 | 0.64 | 1.11 | 1.65 | 149 |
| Total ions | 19% | - | 1.89 | 1.80 | 2.71 | 2.95 | 109 |
| Temperature (°C) | 0% | - | - | 8.57 | 8.13 | 8.92 | 110 |
| Relative humidity (%) | 0% | - | - | 86.8 | 84.5 | 12.6 | 15 |
| Wind speed (m/s) | 0% | - | - | 3.63 | 4.37 | 3.09 | 71 |
| Precipitation (mm/day) | 2% | - | - | 0.60 | 4.15 | 9.71 | 234 |

4. Lines 113–114: Is there any specific reason why the authors averaged the original data to daily values? If the original data is hourly or 3-hr, it should be possible to obtain 3-hr, 6-hr or 12-hr averages, which could result in a larger database for PMF and PCA. Isn't it better?

The Hg data were 3-hrs. However, the concentrations of $SO_2$ and $HNO_3$, $PM_{2.5}$ (2009 only), and particulate $SO_4^{2-}$, $NO_3^-$, $Mg^{2+}$, $Cl^-$, $K^+$, $Ca^{2+}$, $NH_4^+$, and $Na^+$ were daily values. Thus, hourly or 3-hr concentrations of GEM, GOM, PBM, $O_3$ and meteorological data were averaged into daily values because PMF and PCA require the same interval for all input variables.

5. Line 145: The expression "resultant PMF result" seems repetitive.

Agree, the word "resultant" has been removed.

6. Lines 192–193: Since the PCA analysis has already been conducted in Cheng et al. (2013), I think the current title of the manuscript is inappropriate. It could give the readers misimpression that this is partially repeated from the previous study. To my understanding, the methodology of

this study is the novelty of this paper. Therefore, it is better to embody the methodology in the title.

Agree, the tile has been revised as "Potential sources and processes affecting speciated atmospheric mercury at Kejimkujik National Park, Canada: comparison of receptor models and data treatment methods".

7. Line 203: Have the authors checked the inter-correlations between any two of the major PCs? Varimax is an orthogonal rotation method, which requires the PCs to be independent on each other. This validation process for the applicability of the Varimax rotation could be mentioned here.

We have checked the correlations between PCs using an oblique rotation method (direct oblimin) instead of Varimax rotation. The correlations between factors in Case 09-C, Case 10-C and Case 10-C&M are all below the Tabachnick and Fiddell threshold of 0.32 (Tabachnick and Fiddell, 2007) indicating the solution remains nearly orthogonal. The correlation between factor combustion/industrial emission and factor Hg wet deposition had a correlation of -0.33. However, the factor loadings between direct oblimin and varimax rotation were not very different. Therefore, we kept the original varimax results and added an explanation of using an oblique rotation method to verify the inter-correlations between the components in the revised paper.

8. Table 5 and Figure 1-2: NO3 in Table 5 should be $NO_3^-$. All the "+" and "-" signs cannot be omitted in Figure 1-2. NO3 and $NO_3^-$ stand for different compounds.

Table 5 and Figures 1-2 have been modified as suggested.

9. Line 212: From the context (Lines267–268), Combustion Emission include both coal combustion and biomass burning? It is better to mention it here. Does open biomass burning or wildfires included in F1?

We agree with the reviewer that it should be clarified early on that combustion emissions include both coal combustion and biomass burning. However there are not enough pollutant markers to distinguish between the various types of combustion sources, such as open biomass burning from wildfires. In the revised manuscript, we have added "Combustion Emission includes fuel combustion and biomass burning. The small contributions of $Ca^+$ (19%) and $K^+$ (22%) suggest a minor impact of biomass burning."

10. Line 232: Can the authors specify what types of sources could be Industrial Sulfur? Non-ferrous metal melting? What could be the possible Industrial Sulfur sources in this region?

Added as suggested, "As shown in Table S1, point sources of industrial sulfur in the province of Nova Scotia include tire production, engineered wood production, food industry, and universities. Coal-fired power plants and metal production are major sources of sulfur; however there are no combustion sources close to the sampling site. These sources are located in eastern

U.S., which could be transported to the site.  Mobile sources of sulfur are ships and vessels from nearby ports (Cheng et al., 2013)."

11. Line 238: The authors mentioned biomass combustion in this part while the name of Factor 3 is Photochemical Process and Re-emission of Hg. Why is it necessary for the biomass combustion to be related to Re-emission? Is it possible that F1 is composed of coal combustion and controlled biomass combustion which are usually mixed from regional sources while F3 is composed of mineral dust and open biomass burning/wildfires which are usually mixed in long-range transport?

We agree with the reviewer the need to clarify. For F1 (Combustion Emission), the following sentences have been added: "Combustion Emission includes fuel combustion and biomass burning.  The small contributions of $Ca^+$ (19%) and $K^+$ (22%) suggest a minor impact of biomass burning." For F3, we have added the following, "An examination of the time series factor profiles revealed that model-reproduced $K^+$, $O_3$ and GEM, GOM, PBM concentrations (in this factor) were rather smooth. The impact of biomass burning seems to be small in this factor due to a lack of high $K^+$, $O_3$, and Hg concentration period or episodes identified.  The relatively stable patterns of $K^+$ and GEM suggest re-emission of GEM while GOM was high in spring with elevated $O_3$, indicating enhanced photochemistry."

12. Table 6: The performance of 2009 GOM and 2010 PBM is poor to me. I don't think the previous discussion linked to these two parts can be validated. Is it possible to improve the model performance by using the 3-hr or 6-hr averages instead of daily averages to increase the size of the database?

In this study, the model performance was evaluated using a number of indexes, include scaled residual plot to evaluate distribution of residuals, Obs/Pred scatter plot to evaluate overall model-measurement agreement, Obs/Pred time series to visualize the model's ability to reproduce monitored concentrations, the Pred/Obs ratios and the annual Predmean/Obsmean ratios to quantify agreement between predicted and observed Hg concentrations on day-to-day and annual basis, respectively. Those analyses indeed suggested that the model performance of 2009 GOM and 2010 PBM was poor.

We agree with the reviewer that it could be possible to improve the model performance by using the 3-hr or 6-hr averages instead of daily averages to increase the size of the database. Unfortunately, the concentrations of $SO_2$ and $HNO_3$, $PM_{2.5}$ (2009 only), and particulate $SO_4^{2-}$, $NO_3^-$, $Mg^{2+}$, $Cl^-$, $K^+$, $Ca^{2+}$, $NH_4^+$, and $Na^+$ were daily values. Thus, hourly $O_3$ and meteorological data, as well as 3-hr concentrations of GEM, GOM, PBM were averaged into daily values because PMF and PCA require the same interval for all input variables.

**References** (papers used in this response but not listed in the reference section of the revised manuscript)

Jaffe, D. A., Lyman, S., Amos, H. M., Gustin, M. S., Huang, J., Selin, N. E., Levin, L., Schure, A. t., Mason, R. P., Talbot, R., Rutter, A., Finley, B., Jaeglé, L., Shah, V., McClure, C., Ambrose, J., Gratz, L., Lindberg, S., Weiss-Penzias, P., Sheu, G.-R., Feddersen, D., Horvat, M.,

Dastoor, A., Hynes, A. J., Mao, H., Sonke, J. E., Slemr, F., Fisher, J. A., Ebinghaus, R., Zhang, Y., and Edwards, G.: Progress on Understanding Atmospheric Mercury Hampered by Uncertain Measurements, Environ. Sci. Technol., 48, 7204-7206, 10.1021/es5026432, 2014.

Jones, C. P., Lyman, S. N., Jaffe, D. A., Allen, T., and O'Neil, T. L.: Detection and quantification of gas-phase oxidized mercury compounds by GC/MS, Atmospheric Measurement Techniques, 9, 2195-2205, 10.5194/amt-9-2195-2016, 2016.

Tabachnick, B. G., and Fidell, L. S.: Using Multivariate Statistics, 5th ed., Pearson Allyn & Bacon, Upper Saddle River, NJ, 2007.

[revised manuscript text omitted]

**Contents**

**Section 1. Selection of the number of PMF factors**

**Section 2. Stability of PMF model runs**

**Table S1**. Point Source eEmissions of Hg and other pollutants reported in NPRI in the province of Nova Scotia within 150 Km of the sampling site (Data source: Environmental Canada, 2016). Bold facilities are shown in Figure 1.

**Table S2**. Coefficients of cross-correlation among all variables in 2009 (bold numbers are significant at $p<0.05$).

**Table S3**. Coefficients of cross-correlation among all variables in 2010 (bold numbers are significant at $p<0.05$).

**Table S4**. PMF factor contributions to speciated Hg and ratios of predicted to observed annual Hg concentrations in 2009.

**Table S5**. PMF factor contributions to speciated Hg and ratios of predicted to observed annual Hg concentrations in 2010.

**Table S6**. Pearson correlation coefficients between Hg forms and other compounds in Case 2009, Case 09+mean, and Case 09+median (bold numbers are significant at $<0.05$).

**Table S7**. Pearson correlation coefficients between Hg forms and other compounds in Case 2010, Case 10+mean, and Case 10+median (bold numbers are significant at $<0.05$).

**Figure S51**. Obs/Pred scatter plot in 2009. a) Case 2009, b) Case 09+mean, c) Case 09+median, d) Case 09+RM, e) Case 09-RM, and f) Case 09ScaleRM, observed GOM and PBM have been scaled.

**Figure S62**. Obs/Pred scatter plot in 2010. a) Case 2010, b) Case 10+mean, c) Case 10+median, d) Case 10+RM, e) Case 10-RM, and f) Case 10ScaleRM, observed GOM and PBM have been scaled.

**Figure S73**. Obs/Pred time series in 2009. a) Case 2009, b) Case 09+mean, c) Case

09+median, d) Case 09+RM, e) Case 09-RM, and f) Case 09ScaleRM, observed GOM and PBM have been scaled.

**Figure S84**. Obs/Pred time series in 2010. a) Case 2010, b) Case 10+mean, c) Case 10+median, d) Case 10+RM, e) Case 10-RM, and f) Case 10ScaleRM, observed GOM and PBM have been scaled.

**Section 1. Selection of the member of PMF factors**

The number of PMF factors needs to be chosen according to the understanding of the sources impacting the samples utilized. When the background information is not enough to determine the number of factors, several methods could be used to determine the range of the number of the factors. The maximum individual column mean (IM) and the maximum individual column standard deviation (IS) of the scaled residual matrix can be used to identify the range of the number of factors. IM and IS will show a drastic drop when the number of factors increase up to a critical value (Lee et al., 1999). The optimal number of factors should be no less than the critical value. The trend of dQ also provides useful information on deciding the number of factors. When dQ becomes small as the number of factor increases, there might be too many factors (Hopke, 2000; Brown et al., 2015). Runs with different numbers of factors in the range determined by IM, IS, and dQ should be conducted. The model performance and the interpretability of factors in each run should be evaluated. The optimal PMF solution should be a compromise of those indexes and the interpretability of the factor profiles and their comparability with those from the literature (Belis et al, 2015a, 2015b; Cesari et al., 2016).

In this study, the IM and IS were calculated to determine the number of factors. The IM and IS dropped dramatically in 2009 when the number of factor increased to 3 (Figure S1). In the line plot of Q(Robust) and Q(true) vs. the number of factors (Figure S2), no significant decreases were found when the number of factors is larger than 5 in 2009. Therefore, the PMF was run using the number of factors from 3 to 5 in 2009. In 2010, the decrease of IS value was gradual while the IM value experienced a drastic drop when the number of factors increased to 3 (Figure S3). The trend of the Q (Robust) and Q (True) in 2010 is similar to 2009 (Figure S4). Therefore, the PMF runs with the number of factors from 3 to 5 were also conducted in 2010. The number of the factors selected (4) is a compromise of the trends of these indexes and the physical meanings of the factors obtained following Cesari et al. (2016). A detailed comparison of the physical meanings of solutions with different number of factors can be found in Liao (2016).

[Figure]

**Figure S1**. IM and IS vs number of PMF factors in 2009.

[Figure]

**Figure S2**. Q(Robust) and Q(true) vs number of PMF factors in 2009.

[Figure]

**Figure S3**. IM and IS vs number of PMF factors in 2010.

[Figure]

**Figure S4**. Q(Robust) and Q(true) vs number of PMF factors in 2010.

**Table S1**. Point source emissions of Hg and other pollutants reported in NPRI within Nova Scotia (EC, 2016). Bold facilities are shown in Figure 1. (updated)

| Facility | Location (lat, long) | Distance to KEJ/direction | Hg (Kg) | | SO$_2$ (Tonnes) | | NO$_2$ (Tonnes) | | NH$_3$ (Tonnes) | |
|---|---|---|---|---|---|---|---|---|---|---|
| | | | 2009 | 2010 | 2009 | 2010 | 2009 | 2010 | 2009 | 2010 |
| **Brooklyn Power** | Brooklyn (44.05°N, 64.70°W) | 50 Km southeast | 0 | 0 | 9.9 | 26 | 309 | 259 | 0 | 0 |
| **Michelin North America (Canada)- Bridgewater Plant** | Bridgewater (44.39°N, 64.54°W) | 53 Km east | 0 | 0 | 195 | 184 | 68 | 63 | 0 | 0 |
| High Liner Foods Inc. | Lunenburg (44.37°N,64.30°W) | 72 Km east | 0 | 0 | 27 | 27 | 0 | 0 | 0 | 0 |
| Department of National Defence – 14 Wing Greenwood | Greenwood (44.98°N, 64.91°W) | 75 Km north | 0 | 0 | 55 | 68 | 19 | 18 | 0 | 0 |
| **Louisana Pacific Canada Ltd.** | East River (44.58°N, 64.16°W) | 88 Km northeast | 0 | 0 | 122 | 102 | 100 | 99 | 0 | 0 |
| Maple Leaf Foods – Larsen Packers Limited | Berwick (45.05°N, 64.75°W) | 89 Km northeast | 0 | 0 | 51 | 38 | 0 | 0 | 0 | 0 |
| **Michelin North America (Canada) - Waterville Plant** | Waterville (45.05°N, 64.65°W) | 92 Km northeast | 0 | 0 | 162 | 182 | 57 | 62 | 0 | 0 |
| Acadia University – Acadia Campus | Wolfville (45.08°N, 64.37°W) | 108 Km northeast | 0 | 0 | 77 | 73 | 27 | 26 | 0 | 0 |
| CKF. Inc. | Hantsport (45.06°N, 64.17°W) | 116 Km northeast | 0 | 0 | 66 | 57 | 21 | 72 | 0 | 0 |
| **Minas Basin Pulp and Power** | Hantsport (45.07°N, 64.17°W) | 116 Km northeast | 0 | 0 | 225 | 260 | 66 | 76 | 0 | 0 |
| Mount Saint Vincent University | Halifax (44.67°N, 63.65°W) | 129 Km northeast | 0 | 0 | 27 | 13 | 7.2 | 3.9 | 0 | 0 |
| Department of National Defence – Canadian Forces Ammunition Depot | Bedford (44.71°N, 63.63°W) | 131 Km northeast | 0 | 0 | 56 | 50 | 0 | 0 | 0 | 0 |

**Table S1** – Continued 1

| Facility | Location (lat, long) | Distance to KEJ/direction | Hg (Kg) | | SO$_2$ (Tonnes) | | NO$_2$ (Tonnes) | | NH$_3$ (Tonnes) | |
|---|---|---|---|---|---|---|---|---|---|---|
| | | | 2009 | 2010 | 2009 | 2010 | 2009 | 2010 | 2009 | 2010 |
| Department of National Defence - Windsor Park | Halifax (44.66°N, 63.61°W) | 132 Km northeast | 0 | 0 | 59 | 44 | 36 | 30 | 0 | 0 |
| **Department of National Defence – Stadacona/Dockyard** | Halifax (44.66°N, 63.58°W) | 133 Km northeast | 0 | 0 | 211 | 177 | 58 | 51 | 0 | 0 |
| Capital Health – Camp Hill Site Heating Plant | Halifax (44.64°N, 63.59°W) | 133 Km northeast | 0 | 0 | 15 | 12 | 14 | 20 | 0 | 0 |
| **DalHousie University** | Halifax (44.64°N, 63.59°W) | 133 Km northeast | 0.18 | 0.15 | 253 | 260 | 89 | 72 | 0 | 0 |
| Saint Mary's University | Halifax (44.63°N, 63.58°W) | 133 Km northeast | 0 | 0 | 1.2 | 0 | 3 | 0 | 0 | 0 |
| Oland Brewery | Halifax (44.66°N, 63.60°W) | 133 Km Northeast | 0 | 0 | 31 | 0 | 0 | 0 | 0 | 0 |
| **Nova Scotia Power – Tufts Cove Generating Station** | Dartmouth (44.67°N, 63.60°W) | 134 Km northeast | 0 | 0 | 2,205 | 2,205 | 3,054 | 3,054 | 0 | 0 |
| **Capital Health-Victoria General Hospital Central Heating Plant** | Halifax (44.64°N, 63.58°W) | 134 Km northeast | 0 | 0 | 215 | 7.6 | 60 | 19 | 0 | 0 |
| Maritime Paper Products Ltd. | Dartmouth (44.70°N, 63.60°W) | 134 Km northeast | 0 | 0 | 7.2 | 0.868 | 3.1 | 2.1 | 0 | 0 |
| Nova Scotia Power –Burnside Combustion Turbines | Dartmouth (44.71°N, 63.61°W) | 134 Km northeast | 0 | 0 | 0 | 0 | 60 | 40 | 0 | 0 |
| Capital Health – Nova Scotia Hospital Central Heating Plant | Dartmouth (44.65°N, 63.55°W) | 136 Km northeast | 0 | 0 | 3.3 | 1.1 | 9.3 | 8.7 | 0 | 0 |
| **Imperial Oil – Dartmouth Refinery** | Dartmouth (44.64°N, 63.54°W) | 137 Km northeast | 2.6 | 2.9 | 4,231 | 3,073 | 1,543 | 1,251 | 0.593 | 2.2 |
| **Department of National Defence – 12 Wing Shearwater** | Shearwater (44.63°N, 63.51°W) | 138 Km northeast | 0 | 0 | 150 | 127 | 43 | 38 | 0 | 0 |
| Martells Contracting | Elmsdale (44.96°N, 63.48°W) | 154 Km northeast | 0 | 0 | 28 | 17 | 4.5 | 2.8 | 0 | 0 |

**Table S1** – Continued 2

| Facility | Location (lat, long) | Distance to KEJ/direction | Hg (Kg) | | SO₂ (Tonnes) | | NO₂ (Tonnes) | | NH₃ (Tonnes) | |
|---|---|---|---|---|---|---|---|---|---|---|
| | | | 2009 | 2010 | 2009 | 2010 | 2009 | 2010 | 2009 | 2010 |
| The Shaw Group Ltd. | Hardwoodlands (45.07°N, 63.52°W) | 160 Km northeast | 0 | 0 | 0 | 0 | 27 | 19 | 0 | 0 |
| **Lafarge Canada Inc. – Brookfield Cement Plant** | Brookfield (45.24°N, 63.33°W) | 180 Km northeast | 5 | 5.9 | 562 | 667 | 498 | 591 | 0 | 0 |
| Polycello | Amherst (45.82°N, 64.23°W) | 183 Km northeast | 0 | 0 | 0.003 | 0.002 | 0.462 | 0.335 | 0 | 0 |
| Enligna Canada Inc. | Middle Musquodoboit (45.13°N, 62.95°W) | 188 Km northeast | 0 | 0 | 2.8 | 2.9 | 25 | 26 | 0 | 0 |
| Oxford Frozen Foods | Oxford (45.73°N, 63.85°W) | 188 Km northeast | 0 | 0 | 66 | 59 | 0 | 0 | 0.9 | 0 |
| Municipality of the county of Colchester – Wastewater Treatment Facility | Truro (45.37°N, 63.34°W) | 188 Km northeast | 0 | 0 | 0 | 0 | 0 | 0 | 2 | 0.08 |
| Crossley Carpet Mills Limited | Truro (45.35°N, 63.29°W) | 189 Km northeast | 0 | 0 | 40 | 32 | 12 | 11 | 0 | 0 |
| Rothsay | Truro (45.36°N, 63.31°W) | 189 Km northeast | 0 | 0 | 77 | 60 | 0 | 0 | 0 | 0 |
| Stanfield's Ltd. | Truro (45.37°N, 63.28°W) | 191 Km northeast | 0 | 0 | 21 | 21 | 0 | 0 | 0 | 0 |
| Stella-Jones Inc. | Truro (45.38°N, 63.27°W) | 192 Km northeast | 0 | 0 | 12 | 19 | 2.9 | 4.2 | 0 | 0 |
| **The Canadian Salt Company Limited – Pugwash Mine and Refinery** | Pugwash (45.84°N, 63.66°W) | 209 Km northeast | 0 | 0 | 168 | 153 | 32 | 31 | 0 | 0 |
| **Michelin North America (Canada) – Pictou County Plant** | New Glasgow (45.62°N, 62.74°W) | 245 Km northeast | 0 | 0 | 209 | 229 | 72 | 78 | 0 | 0 |
| Maritime Steel and Foundries Limited | New Glasgow (45.58°N, 62.64°W) | 245 Km northeast | 0 | 0 | 0.25 | 0 | 0.875 | 0 | 0 | 0 |

**Table S1** – Continued 3

| Facility | Location (lat, long) | Distance to KEJ/direction | Hg (Kg) | | SO$_2$ (Tonnes) | | NO$_2$ (Tonnes) | | NH$_3$ (Tonnes) | |
|---|---|---|---|---|---|---|---|---|---|---|
| | | | 2009 | 2010 | 2009 | 2010 | 2009 | 2010 | 2009 | 2010 |
| **Nova Scotia Power – Trenton Generating Station** | Trenton (45.62°N, 62.64°W) | 248 Km northeast | 33 | 19 | 30,429 | 19,257 | 5,126 | 5,577 | 0 | 0 |
| Nova Forge Corporation | Trenton (45.62°N, 62.64°W) | 248 Km northeast | 0 | 0 | 3.1 | 0 | 0 | 0 | 0 | 0 |
| **Northern Pulp Nova Scotia Corporation** | New Glasgow (45.65°N, 62.72°W) | 266 Km northeast | 0 | 0 | 246 | 89 | 688 | 676 | 42 | 46 |
| St. Francis Xavier University | Antigonish (45.62°N, 61.99°W) | 291 Km northwast | 0 | 0 | 41 | 36 | 25 | 17 | 0 | 0 |
| **Exxonmobil Canada Properties – Goldboro Gas Plant** | Goldboro (45.17°N, 61.61°W) | 300 Km northeast | 0 | 0 | 0 | 0 | 521 | 415 | 0 | 0 |
| **Nova Scotia Power – Point Tupper Generating Station** | Port Hawkesbury (45.58°N, 61.35°W) | 335 Km northeast | 12 | 9.5 | 9,394 | 5,721 | 1,952 | 1,952 | 0 | 0 |
| **Newpage Port Hawkesbury Corp.** | Port Hawkesbury (45.60°N, 61.36°W) | 355 Km northeast | 0 | 0 | 294 | 85 | 404 | 306 | 0.23 | 0.23 |
| Exxonmobil Canada Properties – Point Tupper Fractionation Plant | Port Hawkesbury (45.58°N, 61.34°W) | 335 Km northeast | 0 | 0 | 0 | 0 | 48 | 23 | 0 | 0 |
| **Exxonmobil Canada Properties – Thebaud Platform** | Offshore (43.01°N, 59.98°W) | 402 Km east | 0 | 0 | 0 | 0 | 135 | 126 | 0 | 0 |
| Exxonmobil Canada Properties – North Triumph Platform | North Triumph Platform (43.01°N, 58.98°W) | 433 Km east | 0 | 0 | 0 | 0 | 26 | 29 | 0 | 0 |
| **Nova Scotia Power – Point Aconi Generating Station** | Point Aconi (46.32°N, 60.30°W) | 442 Km northeast | 2.7 | 2.8 | 3,627 | 3,365 | 1,759 | 1,747 | 0 | 0 |
| Exxonmobil Canada Properties – Venture Platform | Venture Platform (44.06°N, 59.58°W) | 450 Km east | 0 | 0 | 18 | 0 | 54 | 51 | 0 | 0 |

**Table S1** – Continued 4

[revised manuscript text omitted]